# AgraSSt: Approximate Graph Stein Statistics for Interpretable Assessment of Implicit Graph Generators

**Wenkai Xu**
Department of Statistics
University of Oxford
wenkai.xu@stats.ox.ac.uk

**Gesine Reinert**
Department of Statistics
University of Oxford
and Alan Turing Institute
reinert@stats.ox.ac.uk

## Abstract

We propose and analyse a novel statistical procedure, coined *AgraSSt*, to assess the quality of graph generators which may not be available in explicit forms. In particular, AgraSSt can be used to determine whether a *learned* graph generating process is capable of generating graphs which resemble a given input graph. Inspired by Stein operators for random graphs, the key idea of AgraSSt is the construction of a kernel discrepancy based on an operator obtained from the graph generator. AgraSSt can provide interpretable criticisms for a graph generator training procedure and help identify reliable sample batches for downstream tasks. We give theoretical guarantees for a broad class of random graph models. Moreover, we provide empirical results on both synthetic input graphs with *known* graph generation procedures, and real-world input graphs that the state-of-the-art (deep) generative models for graphs are *trained* on.

## 1 Introduction

Generative models for graphs have received increasing attention in the statistics and machine learning communities. Recently, deep neural networks have been utilised to learn rich representations from graph structures and generate graphs [Dai et al., 2020, Li et al., 2018, Liao et al., 2019, You et al., 2018]. However, due to the often opaque deep learning procedures, these deep generative models are usually implicit, which hinders theoretical analysis to assess how close the generated samples are in their distributional properties to the graph distribution they are meant to be sampled from.

Learning instead parametric models from an explicit pre-specified probability distribution class would allow to use the learned parameters for model assessment. However, parametric models may only capture a fraction of the graph features and have restricted modelling power. Parameter estimation can be inconsistent [Shalizi and Rinaldo, 2013] even for well-specified models, and may lead to wrong conclusions in model assessments. Using instead deep generative models for graphs may surpass some of these issues by learning rich graph representations, but methods to assess the quality of such implicit graph generators are lacking. In principle, nonparametric hypothesis tests can be useful to assess complex models, such as kernel-based tests procedures that utilise functions in a reproducing kernel Hilbert space (RKHS) [Berlinet and Thomas, 2004]. When the models are described in the form of explicit probabilities, goodness-of-fit tests [Chwialkowski et al., 2016, Liu et al., 2016] may apply; however, goodness-of-fit testing procedures are generally not applicable for implicit models. Instead, if a large set of samples from the target distribution is observed, one may generate samples from the implicit model and perform a two-sample test such as a maximum mean discrepancy (MMD) test [Gretton et al., 2007] for model assessment [Jitkrittum et al., 2017, Xu and Matsuda, 2020, 2021].

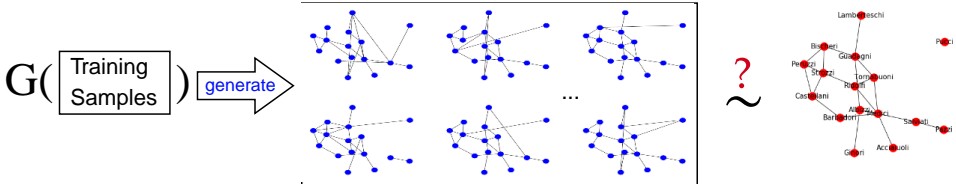

Figure 1: Illustration for the assessment task: a graph generator $G$, which is learned from training samples, generates a set of network samples (vertices in blue), and is assessed against the target graph, which here is Padgett's Florentine marriage network (with vertices in red labelled as family names).

Yet, in real-world applications, often only a single graph from the target distribution is observed [Bresler and Nagaraj, 2018, Reinert and Ross, 2019]. In this case, the MMD methods for model assessment via two-sample testing procedures cannot be used to assess implicit graph generators. To our knowledge, beyond simple Monte Carlo tests with often poor power, no principled test is available for assessing the quality of implicit graph generators. Figure.1 illustrates the task: a given graph generator $G$ which is learned from a set of training samples can generate samples of user-defined size. The task is to assess whether $G$ can generate samples from the same distribution that generates the observed graph, for example, Padgett's Florentine marriage network [Padgett and Ansell, 1993]. The samples in Figure.1 are generated by a Cross-Entropy Low-rank Logit (CELL) model [Rendsburg et al., 2020] trained on Padgett's Florentine marriage network, but any graph generator could be used.

This paper makes three main contributions. (1) We introduce Approximate graph Stein Statistics (AgraSSt) in Section 3, which opens up a *principled way* to understand implicit graph generators. AgraSSt is a variant of a kernel Stein discrepancy based on an empirical Stein operator for conditional distributions of general random graph models. The testing procedure is inspired by gKSS, a goodness-of-fit testing procedure for explicit exponential random graph models (ERGMs) [Xu and Reinert, 2021]. (2) We provide theoretical guarantees for AgraSSt (Section 3). (3) We propose interpretable model criticisms when there is a model misfit, and we identify representative synthetic sample batches when the model is a good fit (Section 4). Further, in Section 2 we review gKSS. We provide empirical results in Section 5 and a discussion in Section 6. Proof details, more background, theoretical and empirical results, and implementation details are found in the Supplementary Information (SI).The code for the experiments is available at `https://github.com/wenkaixl/agrasst`.

## 2 Background: graph kernel Stein statistics

In [Xu and Reinert, 2021], a goodness-of-fit testing procedure called gKSS for exponential random graph models is introduced, which assumes that only a single network may be observed in the sample. As AgraSSt is inspired by gKSS, and gKSS serves as comparison method when the observed network is known to be generated from an exponential random graph model (ERGM), we briefly review gKSS. The notation introduced in this section is used throughout the paper.

ERGMs are extensively used for social network analysis [Wasserman and Faust, 1994, Holland and Leinhardt, 1981, Frank and Strauss, 1986]; a special case are Bernoulli random graphs. ERGMs model random graphs via a Gibbs measure with respect to a chosen set of network statistics, such as number of edges, 2-stars, and triangles. Denote by $\mathcal{G}_n^{lab}$, the set of vertex-labeled graphs on $n$ vertices, with $N = n(n-1)/2$ possible undirected edges. Encode $x \in \mathcal{G}_n^{lab}$ by an ordered collection of $\{0,1\}$-valued variables $x = (x^{(ij)})_{1 \le i < j \le n} \in \{0,1\}^N$ where $x^{(ij)} = 1$ if and only if there is an edge between $i$ and $j$. We denote an (ordered) vertex-pair index $s = (i,j)$ by $s \in [N] := \{1, \ldots, N\}$. Let $t_1(x)$ count the number of edges in $x$ and let $t_2(x), \ldots, t_k(x)$ count other small graphs on at most $n$ vertices in $x$ (possibly scaled; for details including a precise definition, Definition A.3, see SI A.2). For $\beta = (\beta_1, \ldots, \beta_k)^\top \in \mathbb{R}^k$ and $t(x) = (t_1(x), \ldots, t_k(x))^\top \in \mathbb{R}^k$ we say that $X \in \mathcal{G}_n^{lab}$ follows the exponential random graph model $X \sim \mathrm{ERGM}(\beta, t)$ if for $\forall x \in \mathcal{G}_n^{lab}$,

$$q(X = x) = \frac{1}{\kappa_n(\beta)} \exp\left(\sum_{l=1}^{k} \beta_l t_l(x)\right). \tag{1}$$

Here $\kappa_n(\beta)$ is a normalisation constant. In this model, $t_\ell, \ell = 1, \ldots, k$, are sufficient statistics. Parameter estimation $\hat{\beta}_l$ for $\beta_l$ is *only* possible when the sufficient statistics are specified a priori; see SI.B.1 for estimation details. In modern graph learning procedures, e.g. deep generative learning, the sufficient statistics $t_l$ of Eq.(1) may not be obtained explicitly.

In [Reinert and Ross, 2019] the exponential random graph distribution in Eq. (1) is characterised by a so-called *Stein operator*, as follows. Let $e_s \in \{0, 1\}^N$ be a vector with 1 in coordinate $s$ and 0 in all others; $x^{(s,1)} = x + (1 - x_s)e_s$ has the $s$-entry replaced of $x$ by the value 1, and $x^{(s,0)} = x - x_s e_s$ has the $s$-entry of $x$ replaced by the value 0; moreover, $x_{-s}$ is the set of edge indicators with entry $s$ removed. For a function $h : \{0, 1\}^N \to \mathbb{R}$, let $\Delta_s h(x) = h(x^{(s,1)}) - h(x^{(s,0)})$. Set $q_X(x^{(s,1)}|x_{-s}) = \mathbb{P}(X^s = 1|X_{-s} = x_{-s})$. Define the operator

$$\mathcal{A}_{\beta,t}f(x) = \frac{1}{N} \sum_{s \in [N]} \mathcal{A}_q^{(s)} f(x), \quad \mathcal{A}_q^{(s)} f(x) = q(x^{(s,1)}|x_{-s})\Delta_s f(x) + \left( f(x^{(s,0)}) - f(x) \right). \quad (2)$$

Then under mild conditions Reinert and Ross [2019] show that if $\mathbb{E}_p[\mathcal{A}_{\beta,t}f] = 0$ for all smooth test functions $f$, then $p$ must be the distribution of $\mathrm{ERGM}(\beta, t)$. Thus, this operator characterises $\mathrm{ERGM}(\beta, t)$. For the derivation of AgraSSt it is of interest to see how this operator is obtained. It is indeed the generator of a so-called *Glauber* Markov chain on $\mathcal{G}_n^{lab}$ with transition probabilities

$$\mathbb{P}(x \to x^{(s,1)}) = N^{-1} - \mathbb{P}(x \to x^{(s,0)}) = N^{-1}q_X(x^{(s,1)}|x).$$

With the ERGM Stein operator in Eq.(2) and a rich-enough RKHS test function class $\mathcal{H}$, Xu and Reinert [2021] propose a graph kernel Stein statistics (gKSS) to perform goodness-of-fit testing on an *explicit* ERGM when a single network sample is observed. With the summand components in Eq.(2), the Stein operator can be seen as taking expectation over vertex-pair variables $S \in [N]$ with uniform probability $\mathbb{P}(S = s) \equiv N^{-1}$ independently of $x$, namely

$$\mathcal{A}_q f(x) = \sum_{s \in [N]} \mathbb{P}(S = s)\mathcal{A}_q^{(s)} f(x) =: \mathbb{E}_S[\mathcal{A}_q^{(S)} f(x)].$$

For a fixed graph $x$, gKSS is defined as

$$\mathrm{gKSS}(q; x) = \sup_{\|f\|_{\mathcal{H}} \leq 1} \left| \mathbb{E}_S[\mathcal{A}_q^{(S)} f(x)] \right|, \quad (3)$$

where the function $f$ is chosen to best distinguish $q$ from $x$. For an RKHS $\mathcal{H}$ associated with kernel $K$, by the reproducing property of $\mathcal{H}$, the squared version of gKSS admits a quadratic form representation $\mathrm{gKSS}^2(q; x) = \langle \mathbb{E}_S[\mathcal{A}_q^{(S)} K(x, \cdot)], \mathbb{E}_S[\mathcal{A}_q^{(S)} K(x, \cdot)] \rangle$, which can be computed readily. More background can be found in SI B.2.

## 3   AgraSSt: Approximate Graph Stein Statistic

An implicit graph generator may not admit a probability distribution in the form of Eq.(1). However, the idea of constructing Stein operators based on Glauber dynamics using conditional probability distribution for ERGM serves as a motivation for us to propose Stein operators for conditional graph distributions, to facilitate an (approximate) characterisation for implicit random graph models.

### 3.1   Stein operators for conditional graph distributions

Let $q(x) = \mathbb{P}(X = x)$ be any distribution with support $\mathcal{G}_n^{lab}$. Let $t(x)$ denote a statistic on graphs which takes on finitely many values $\underline{k}$ and let $q_{\underline{k}}(x) = \mathbb{P}(X = x|t(x) = \underline{k})$. We assume that $q_{\underline{k}}(x) > 0$ for all $\underline{k}$ under consideration. For a generic outcome we write $q_t$. We introduce a Markov chain on $\mathcal{G}_n^{lab}$ which transitions from $x$ to $x^{(s,1)}$ with probability

$$q(x^{(s,1)}|t(x_{-s})) = \mathbb{P}(X^s = 1|t(x_{-s})) =: q_t(x^{(s,1)}), \quad (4)$$

and which transitions from $x$ to $x^{(s,0)}$ with probability $q(x^{(s,0)}|t(x_{-s})) = 1 - q_t(x^{(s,1)})$; no other transitions occur. Let

$$\mathcal{A}_{q,t}^{(s)} f(x) = q_t(x^{(s,1)})f(x^{(s,1)}) + q_t(x^{(s,0)})f(x^{(s,0)}) - f(x). \quad (5)$$

---
**Algorithm 1** Estimating the conditional probability
---
**Input:** Graph generator $G$; statistics $t(x)$;
**Procedure:**
 1: Generate samples $\{x_1, \ldots, x_L\}$ from $G$.
 2: For $s \in [N], i \in [n]$, let $n_{s,k}$ be the number of graphs $x_l, l \in [L]$, in which $s$ is present and $t(x_{-s}) = k$.
 3: Estimate the conditional probability of the edge $s$ being present conditional on $t(x_{-s}) = k$ by an estimator $\widehat{g}_t(s; k)$, using a look-up table or smoothing.
**Output:** $\widehat{g}_t(s; k)$ that estimates $q(x^{(s)} = 1 | t(x_{-s}) = k)$.
---

For an ERGM, $t(x)$ could be taken as a sufficient vector of statistics[1], but here we do not assume a parametric network model $q(x)$, and $t(x)$ does not have to be sufficient statistics for $q(x)$.

Recall that an operator is a Stein operator for a distribution $\mu$ if its expectation under $\mu$ is zero. The following result provides a theoretical foundation for AgraSSt and is proven in SI.A.

**Lemma 3.1.** *If $q_{\underline{k}}(x) > 0$ for all $\underline{k}$, then $\mathcal{A}_{q,t=\underline{k}}^{(s)}$ is a Stein operator for the conditional distribution of $X$ given $t(X) = \underline{k}$, and $\sum_s \mathcal{A}_{q,t=\underline{k}}^{(s)}$ is a Stein operator for the conditional distribution of $X$ given $t(X) = \underline{k}$.*

In particular, $\mathbb{E}_{q_{t=\underline{k}}}[\mathcal{A}_{q,t=\underline{k}}^{(s)}] = 0$. Intuitively, if the distribution of $\tilde{X}$ is close to that of $X$, then with $\tilde{Y}_{t=\underline{k}}$ denoting the corresponding random graph with distribution that of $\tilde{X}$ given $t(\tilde{X}) = \underline{k}$, it should hold that $\mathbb{E}[\mathcal{A}_{q,t=\underline{k}}^{(s)}(\tilde{Y}_{t=\underline{k}})] \approx 0$. In this way the Stein operator in Eq.(5) can be used to assess similarity between distributions.

## 3.2 Approximate Stein operators

For implicit models and graph generators $G$, the Stein operator $\mathcal{A}_{q,t}^{(s)}$ in Eq.(5) cannot be obtained without explicit knowledge of $q_t(x^{(s,1)})$. However, given a large number for samples from the graph generator $G$, the conditional edge probabilities $q_t(x^{(s,1)})$ can be estimated. Here we denote by $\widehat{q}_t(x^{(s,1)})$ an estimate of $q_t(x^{(s,1)})$; some estimators will be suggested in Section 3.3.

AgraSSt performs model assessment using an operator which approximates the Stein operator $\mathcal{A}_{q,t}^{(s)}$. We define the approximate Stein operator for the conditional random graph by

$$\mathcal{A}_{\widehat{q},t}^{(s)} f(x) = \widehat{q}_t(x^{(s,1)}) f(x^{(s,1)}) + \widehat{q}_t(x^{(s,0)}) f(x^{(s,0)}) - f(x). \tag{6}$$

The vertex-pair averaged approximate Stein operator is

$$\mathcal{A}_{\widehat{q},t} f(x) = \frac{1}{N} \sum_{s \in [N]} \mathcal{A}_{\widehat{q},t}^{(s)} f(x). \tag{7}$$

## 3.3 Estimation with chosen statistics on graphs

Using the Stein operator for conditional graph distributions, we can obtain the approximate Stein operators in Eq. (6) and Eq. (7) for an implicit graph generator $G$ by estimating $q_t(x^{(s,1)})$. Here $t(x)$ are user-defined statistics. In principle, any multivariate statistic $t(x)$ can be used in this formalism. However, estimating the conditional probabilities using relative frequencies can be computationally prohibitive when the graphs are very large and specific frequencies are rarely observed. Instead, here we consider simple summary statistics, such as edge density, degree statistics or the number of neighbours connected to both vertices of $s$. The estimation procedure is presented in Algorithm 1.

To estimate $q_t(x^{(s,1)})$ in Step 3 of Algorithm 1, a look-up table can be used: If $t(x)$ is a possibly multivariate statistic with a discrete number of outcomes, generate many independent copies from the

---
[1] When conditioning on the sufficient statistics, for ERGMs the resulting Stein operator allows to establish elegant approximation results [Bresler and Nagaraj, 2019, Reinert and Ross, 2019].

synthetic graph generator, and count $n(\underline{k}, s)$, the number of times that vertex-pair $s$ is present in the simulated graphs and $t(x_{-s}) = \underline{k}$. Consistency follows from the law of large numbers. When using smoothing to estimate the conditional probability, consistency will depend on the smoothing method. If the underlying graph has exchangeable edge indicators then $q_t(x^{(s,1)})$ does not depend on the choice of vertex-pair $s$. Then we set set $n(\underline{k}) = \sum_s n(\underline{k}, s)$, and $N_{\underline{k}} = \sum_{i=1}^{L} \sum_s \mathbf{1}(t(x_{-s}) = k)$; if $t = \underline{k}$ we estimate $\widehat{q}_t(x^{(s,1)})$ by

$$\hat{g}_t(\underline{k}) = \frac{n_k}{N_{\underline{k}}} \mathbb{1}(N_{\underline{k}} \geq 1). \tag{8}$$

If $k_{min}, k_{max}$ denote the minimum and maximum values of statistics from simulated graph samples, then for $\underline{k}$ outside this set, the lookup table estimator Eq.(8) is set to estimate $\hat{g}(\underline{k}) = 0$ .

If the underlying graph cannot be assumed to have exchangeable edge indicators or if the statistic $t$ is high dimensional, then any particular $n(\underline{k}, s)$ may not be observed very often. In such a situation we can learn $\hat{g}_t(s; k)$ using kernel ridge regression so that the conditional probabilities for similar $t(x_{-s})$ are predicted in a smooth manner. Estimating $q(x^s | t(x_{-s}) \leq k)$ instead of $q(x^s | t(x_{-s}) = k)$ may provide an alternative, smoother estimate for the conditional probabilities.

The next result shows that the approximate Stein operator achieves the Stein identity asymptotically. For this result, which is proved in SI.A, we use the notation $\|\Delta f\| = \sup_{s \in [N], x} \|\Delta_s f(x)\|$.

**Theorem 3.2.** *Assume $\widehat{q}_t(x^{(s,1)})$ is a consistent estimator for $q_t(x^{(s,1)})$ as $L \to \infty$. Then for any $f$ such that $\|\Delta f\| < \infty$ we have $\mathbb{E}_q[\mathcal{A}_{\widehat{q},t} f(x)] \to \mathbb{E}_q[\mathcal{A}_{q,t} f(x)] = 0$ as $L \to \infty$.*

Proofs and additional analysis are included in SI A. Proposition A.1 in SI A.1 addresses consistent estimation of $\widehat{q}_t$ and SI A.2 provides refined results for ERGMs, including a Gaussian approximation.

## 3.4 AgraSSt for implicit graph generators

The estimated conditional probabilities give an approximate Stein operator for Eq.(5). With the appropriately defined Stein operator from an implicit model given in Eq.(6), we can define AgraSSt, a kernel-based statistic analogous to gKSS in Eq.(3), as

$$\text{AgraSSt}(\widehat{q}, t; x) = \sup_{\|f\|_{\mathcal{H}} \leq 1} \left| N^{-1} \sum_s \mathcal{A}_{\widehat{q},t}^{(s)} f(x) \right|.$$

In SI.A we prove the following result for *edge-exchangeable graphs*, that is, graphs in which all permutations of the edge indicators have the same distribution.

**Theorem 3.3.** *If the graph is edge-exchangeable, then $\text{AgraSSt}^2(\widehat{q}, t; x)$ is a consistent estimator of*

$$\text{gKSS}^2(q; x) = N^{-2} \sum_{s, s' \in [N]} \left\langle \mathcal{A}_q^{(s)} K(x, \cdot), \mathcal{A}_q^{(s')} K(\cdot, x) \right\rangle_{\mathcal{H}}. \tag{9}$$

**Re-sampling Stein statistic** A computationally efficient operator for large $N$ can be derived via re-sampling $B$ vertex-pairs $s_b$, $b = 1, \ldots, B$, from $\{1, \ldots, N\}$, chosen uniformly with replacement, independent of each other and of $x$. This procedure gives rise to a randomised operator; this re-sampled operator is

$$\widehat{\mathcal{A}}_{\widehat{q},t}^B f(x) = \frac{1}{B} \sum_{b \in [B]} \mathcal{A}_{\widehat{q},t}^{(s_b)} f(x).$$

where the expectation of $\widehat{\mathcal{A}}_{\widehat{q},t}^B f(x)$ with respect to re-sampling is

$$\mathbb{E}_B[\widehat{\mathcal{A}}_{\widehat{q},t}^B f(x)] = \mathbb{E}_S[\mathcal{A}_{\widehat{q},t}^{(S)} f(x)] = \mathcal{A}_{\widehat{q},t} f(x).$$

The corresponding re-sampled AgraSSt statistic is

$$\widehat{\text{AgraSSt}}(\widehat{q}, t; x) = \sup_{\|f\|_{\mathcal{H}} \leq 1} \left| \frac{1}{B} \sum_{b \in [B]} \mathcal{A}_{\widehat{q},t}^{(s_b)} f(x) \right|.$$

**Algorithm 2** Assessment procedures for graph generators

---

**Input:** Observed graph $x$; graph generator $G$ and generated sample size $L$; estimation statistic $t$;
    RKHS kernel $K$; re-sampling size $B$; number of simulated graphs $m$; confidence level $\alpha$;

**Procedure:**
1: Estimate $\widehat{q}(x^{(s)}|t(x_{-s}))$ based on Algorithm 1.
2: Uniformly generate re-sampling index $\{s_1, \ldots, s_B\}$ from $[N]$ with replacement.
3: Compute $\tau = \widehat{\text{AgraSSt}}^2(\widehat{q}; x)$ in Eq.(10).
4: Simulate $\{z'_1, \ldots, z'_m\}$ from $G$.
5: Compute $\tau_i = \widehat{\text{AgraSSt}}^2(\widehat{q}; z'_i)$ in Eq.(10).
6: Estimate empirical quantile $\gamma_{1-\alpha}$ via $\{\tau_1, \ldots, \tau_m\}$.

**Output:** Reject the null if $\tau > \gamma_{1-\alpha}$; otherwise do not reject.

---

Similar to Eq.(9), the squared version of $\widehat{\text{AgraSSt}}$ admits a representation in a quadratic form,

$$\widehat{\text{AgraSSt}}^2(\widehat{q}, t; x) = B^{-2} \sum_{b,b' \in [B]} \widehat{h}_x(s_b, s_{b'}), \tag{10}$$

where $\widehat{h}_x(s, s') = \langle \mathcal{A}_{\widehat{q},t}^{(s)} K(x, \cdot), \mathcal{A}_{\widehat{q},t}^{(s')} K(\cdot, x) \rangle_{\mathcal{H}}$. We note that the randomised operator obtained via re-sampling is a form of stochastic Stein discrepancy as introduced in Gorham et al. [2020].

For fixed $x$, under mild conditions the consistency of $\widehat{\text{AgraSSt}}^2(\widehat{q}, t; x)$ as $B \to \infty$ is ensured by the following normal approximation, which follows from Proposition 2 in Xu and Reinert [2021].

**Proposition 3.4.** *Assume that $\widehat{h}_x(s, s')$ in (10) is bounded and that $\widehat{\text{AgraSSt}}(\widehat{q}, t; x)$ has non-zero variance $\sigma^2$. Let $Z$ be a normal variable with mean $\text{AgraSSt}(\widehat{q}, t; x)$ and variance $\sigma^2$. Then there exists an explicitly computable constant $C > 0$ such that for all 3 times continuously differentiable functions $g$ with bounded derivatives up to order 3,*

$$\mathbb{E}[g(\widehat{\text{AgraSSt}}(\widehat{q}, t; x)) - g(Z)] \leq C/B.$$

## 4 Applications of AgraSSt

### 4.1 Assessing graph generators

AgraSSt measures the distributional difference between the underlying distribution of an implicit graph generator $G$ and an observed graph $x$, in order to assess the quality of the generator $G$. The hypothesis testing procedure for the null hypothesis that the observed graph $x$ comes from the same distribution that generates the samples, against the general alternative, is shown in Algorithm 2. We emphasise two features of this procedure. Firstly, for a given generator $G$, AgraSSt directly assesses the quality of the implicit model represented via samples from $G$. Secondly, the generator $G$ can be trained on the observed graph $x$, for example through a deep neural network generator. By learning a deep neural network generator with training samples from the same distribution that generate $x$, AgraSSt can assess the quality of the training procedure, i.e. whether the deep neural network is capable of learning the desired distributions. Additional details are discussed in SI.C.

### 4.2 Interpreting trained graph generators

If the procedure in Algorithm 2 rejects the null hypothesis, the generator may not be suitable for generating samples from the distribution that generates the one observed graph. Hence, understanding where the misfit comes from can be very useful, especially for models trained from black-box deep neural networks. AgraSSt provides an interpretable model criticism by comparing the learned $\widehat{q}_t(x^{(s,1)})$ with the underlying $q_t(x^{(s,1)})$ when available, such as in synthetic experiments from a specified ERGM. Such an interpretation can be also useful to re-calibrate training procedures.

### 4.3 Identifying reliable graph samples

If the procedure in Algorithm 2 does not reject the null hypothesis, there is not enough evidence to reject the hypothesis that the generator is capable of generating graphs that resembles the observed

graph. If a generator $G$ has passed this hurdle then it can be recommended for generating graph samples of the desired type. AgraSSt can also be put to use for the task of sample batch selection. In many scientific studies, only a small batch of representative graph samples may need to be generated for downstream tasks such as privacy-preserving methods where users only access a small number of graph data, or a randomised experimental design for community interaction. To quantify the quality of sample batches via $p$-values : (1) we generate a sample batch of size $m$, say; (2) we perform the steps in **Procedure** from Algorithm 2; (3) we compute the $p$-value $m^{-1}\sum_{i=1}^{m}\mathbb{1}(\tau > \tau_i)$, with $\tau$ as in step 3 and $\{\tau_1, \ldots, \tau_m\}$ as in step 5. If the $p$-value is smaller than a pre-specified threshold, we generate another sample batch; otherwise we accept the current sample batch.

# 5    Empirical results

We first illustrate the performance of AgraSSt on synthetic data, where the null distribution is *known* and we have control of the set-up; in particular we can illustrate the use of AgraSSt for interpretable model criticism. Then we show the performance of AgraSSt on a real-world data application to assess graph generators trained via various deep generative models.

## 5.1    Synthetic experiments

Only few competing approaches are available for our task and many of them are devised specifically for ERGMs. Hence here we use an ERGM, namely the Edge-2Star-Triangle (E2ST) model with

$$q(x) \propto \exp\left(\beta_1 E_d(x) + \beta_2 S_2(x) + \beta_3 T_r(x)\right), \tag{11}$$

where $E_d(x)$ denotes the number of edges of $x$, $S_2(x)$ denotes the 2-Star statistics and $T_r(x)$ denotes the triangle statistics. The parameter vector $\beta = (-2.00, 0.00, 0.01)$ is chosen as the null model, while alternative models are constructed by perturbing the coefficient $\beta_2$ as in Yang et al. [2018], Xu and Reinert [2021]. This particular type of ERGM is chosen because it is the currently most complex ERGM for which a thorough theoretical analysis for parameter estimation is available, see Mukherjee and Xu [2013][2].

### 5.1.1    Related approaches for comparisons

To assess the performance of AgraSSt, we consider the following test statistics which are either tailored or modified to perform assessment for implicit graph generators:

- **Deg** is a degree-based statistics for goodness-of-fit test of exchangeable random graphs [Ouadah et al., 2020] based on the estimated variance of the degree distribution. The statistics can be obtained from empirical degrees from samples generated from the implicit model.

- **TV_deg** denotes the Total-Variation (TV) distance between degree distributions. Hunter et al. [2008] proposes a simulation-based approach to construct graphical goodness-of-fit tests. Xu and Reinert [2021] quantifies this approach using the total-variation distance between the distributions of chosen network statistics; see SI.D.4 for details.

- **MDdeg** is the Mahalanobis distance between degree distributions [Lospinoso and Snijders, 2019].

In the synthetic experiment where the parametric model is known (Eq.(11)), the coefficients $\beta$ are estimated from generated samples for model assessment to provide our baseline approach, denoted by **Param**. Details can be found in SI.B.1. Knowing the explicit null model in the synthetic setting, we can also compute gKSS in Eq.(3), denoted by **Exact** as our benchmark. The Weisfeiler-Lehman graph kernel [Shervashidze et al., 2011] with height parameter 3 is used for kernel-based approaches.

### 5.1.2    Simulation results

The rejection rates for various settings are shown in Figure.2. As the null model is relatively sparse (edge density $11.2\%$), the sparser alternatives are much harder to distinguish while the denser ones are

---

[2]The model also satisfies conditions in Theorem 1.7 in Reinert and Ross [2019] where theoretical properties are studied.

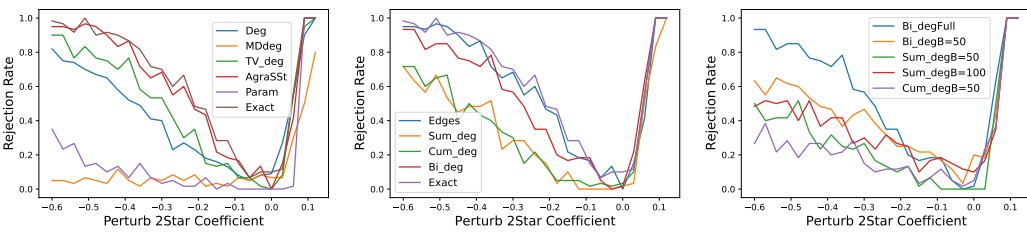

(a) Different assessment approaches    (b) AgraSSt: different estimations    (c) AgraSSt with re-sampling

Figure 2: Synthetic experiment on the E2ST model in Eq.(11) with $\beta_2$ varying on the $x$-axis: 100 trials; $\alpha = 0.05$; $L = 1000$. Abbreviations as in Section 5.1.

easier problems. Figure.2(a) compares the AgraSSt procedure with the approaches from Section 5.1.1; it shows that **AgraSSt** performs competitively to the benchmark **Exact**, which is only available when the model is known explicitly, and outperforms other assessment procedures for implicit models. **TV_deg** is slightly less powerful than **AgraSSt** but outperforms **Deg**. For sparser alternatives (smaller $\beta_2$), **MDdeg** and **Param** have a much lower rejection rate and thus are less powerful. In Figure.2(b), we compare the performance of **AgraSSt** using different estimation methods for $\widehat{q}_t(x^{(s,1)})$; the degree $deg(k)$ of a vertex $k$ is calculated excluding the vertex-pair $s = (i, j)$. The estimation methods are

- **Sum_deg**: for $s = (i, j)$ we set $t = deg(i) + deg(j)$;
- **Cum_deg**: the cumulative distribution function of sum of degrees are used;
- **Bi_deg**: for $s = (i, j)$ the 2-dimensional vector $t = (deg(i), deg(j))$ is used;
- **Edges**: $t$ is the edge density after removing vertex-pair $s$.

From the results, we see that **Edges** outperforms the other estimates, which echos the theoretical results shown in Theorem A.5 in SI, as the coefficient $\beta$ for E2ST satisfies its assumptions. Hence this is the statistics which is used in our real-world experiments, with an exception of SI.D.3. We also see that using both vertex degrees as 2d vector predicts substantially better than using predictors based on sum of degrees of two vertices. In Figure.2(c), the comparison with re-sampling is shown. With increase in re-sampling size, the power of AgraSSt increases.

## 5.2 Real-world applications on deep graph generators

We now assess the performances of a set of state-of-the-art deep generative models for graphs trained on ERGMs and the Karate Club network collected by Zachary [1977]. The Karate Club network has 34 vertices and 78 edges representing friendships. Soon after the data collection the Karate Club separated into two factions. This graph is a benchmark graph for community detection. One would not expect this graph to be close to an $G(n, p)$[3] graph or to be well modelled by an ERGM.

### 5.2.1 Graph generation methods

The graph generation methods to which we apply AgraSSt are the following.

- **GraphRNN** [You et al., 2018] is an architecture to generate graphs from learning two recurrent neural networks (RNN), one a vertex-level RNN and the other an edge-level RNN. The procedure starts from a breadth-first-search for vertex ordering; two RNNs are trained from a sequential procedure.
- **NetGAN** [Bojchevski et al., 2018] utilises an adversarial approach by training an interplay between a generator and a discriminator neural network on graph data.
- **CELL** [Rendsburg et al., 2020] improves on the NetGAN idea by solving a low-rank approximation problem based on a cross-entropy objective.
- **MC** is the standard Monte-Carlo network sampling in the *ergm* suite in R and is used as a baseline when the simulated network is known to follow the model in Eq.(1); $q(x)$ needs to be known.

---

[3]Bernoulli random graph of size $n$, edge probability $p \in [0, 1]$.

### 5.2.2 Generator assessment results

We first train the generative models with samples from ERGMs to assess their ability to generate ERGMs. The test results are shown in Table.1. From the result, we see that for the "reliable" **MC** generator all the assessment statistics presented have well-controlled type-I error. Samples generated from **CELL** deviate not too far from the test level, indicating a good generative model for ERGMs. **NetGAN** and **GraphRNN** both encounter a high rejection rate, implying that the generated samples that are not close to the training E2ST model.

|  | AgraSSt | Deg | MDdeg | TV_deg |
|---|---|---|---|---|
| GraphRNN | 0.42 | 0.02 | 0.04 | 0.27 |
| NetGAN | 0.81 | 0.13 | 0.61 | 0.54 |
| CELL | 0.05 | 0.06 | 0.09 | 0.12 |
| MC | 0.04 | 0.03 | 0.02 | 0.09 |

Table 1: Rejection rates on various assessment approaches, with $L = 1000$; 200 samples to simulate the null; 100 trials; $\alpha = 0.05$. The higher the rejection rate, the worse the model fit. MC is the baseline.

From the density based **AgraSSt**, taking a $G(n, q)$ model, we can interpret the model misfit by checking the estimated $\widehat{q}$. For the true E2ST model used to generate training samples, we have $= 0.112$, while CELL has $\widehat{q} = 0.116$ which is close to the null. GraphRNN estimates $\widehat{q} = 0.128$ which is substantially higher than the null. Although GraphRNN can be powerful in learning local patterns and structures for neighbourhoods [You et al., 2018], here it does not take the overall density sufficiently into account. Due to its limited "look back" and absence of "look forward" on the ordered vertex set during training, the over-generation of edges may have caused this significant difference for learning ERGMs. NetGAN, on the other hand, produces a close estimate $\widehat{q} = 0.106$. However, counting triangles, it only has on average 12.6 triangles, which is far less from the null with expected number of triangles 46.3. NetGAN, due to its random walk adversarial procedure, may not be effective in learning such clustered patterns.

### 5.2.3 Case study: Karate Club network

Next, we assess the performances of these generative models by training on the Karate Club network [Zachary, 1977]. The $p$-values for different testing procedures are shown in Table.2. From the results, we see that AgraSSt rejects samples generated from both GraphRNN and NetGAN trained with the Karate Club network. Although the edge densities generated from the trained GraphRNN (edge density 15.3%)

|  | AgraSSt | Deg | MDdeg | TV_deg |
|---|---|---|---|---|
| GraphRNN | 0.00 | 0.01 | 0.15 | 0.00 |
| NetGAN | 0.00 | 0.02 | 0.59 | 0.00 |
| CELL | 0.34 | 0.09 | 0.17 | 0.61 |

Table 2: $p$-values for models trained on the Karate Club network; 100 samples to simulate the null distribution; rejection at $\alpha = 0.05$ is marked red.

and NetGAN (edge density 13.4%) are comparable with the Karate Club edge density of 13.9%, visual inspection indicates that both GraphRNN and NetGAN samples exhibit a single large component rather than two fairly separated communities in the Karate Club network. This difference is picked up by AgraSSt, Deg and TV_deg, which all reject both models. On the other hand, CELL generates samples that are not rejected by all tests at significance level $\alpha = 0.05$. In Figure.5 in the SI, the Karate Club network is shown in Figure.5(a). Samples from GraphRNN, NetGAN and CELL are shown in Figure.5(b), 5(c) and 5(d) respectively.

SI D includes additional results, including on efficiency, and visualisations. A second case study — the Florentine marriage network from Padgett and Ansell [1993] — is presented in SI D.2; additional visualisations of the reliable sample batch selection procedure described in Section 4.3 are also given.

## 6 Discussions and future directions

In this paper, we propose AgraSSt, a unique general purpose model assessment and criticism procedure for implicit random graph models. As it is based on a kernel Stein statistic, we are able to give theoretical guarantees. AgraSSt not only solves an important problem but also opens up a whole set of follow-up research problems of which we list a few here. (i). Currently AgraSSt is only applied to undirected and unweighted graphs. Extensions to more general graphs as well as to time

series of graphs will be interesting to explore in follow-up work. (ii). AgraSSt could be also helpful to improve design and training of deep graph generative models, e.g. by regularising graph features if there is a misfit. (iii). ERGMs allow for exogenous features to be included in the sufficient statistics. AgraSSt can be based on a variety of statistics $t(x)$; further examples are found in SI D.3. It would also be possible to incorporate exogenous features in the statistics $t(x)$ in AgraSSt, for example using ideas from graph attention networks [Veličković et al., 2018]. Exploring this idea in more detail will be another topic of further research.

As AgraSSt depends on the chosen summary statistic $t(x)$, results have to be interpreted with regards to the respective conditional distributions. Also multiple tests will have the $p$-values to be adjusted to avoid misinterpretation of tests, which could have serious consequences for example in the area of personal health. Investigating the effect of choice of $t(x)$ will be part of future work.

## Acknowledgements

The authors thank the anonymous reviewers for their constructive comments and suggestions to improve the paper. G.R. and W.X. acknowledges the support from EPSRC grant EP/T018445/1. G.R is also supported in part by EPSRC grants EP/W037211/1, EP/V056883/1, and EP/R018472/1.

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
