## A  Proofs and additional theoretical results

### A.1  Proofs

Here we usually omit the $=\underline{k}$ suffix in $\mathcal{A}_{q,t=\underline{k}}^{(s)}$ and write instead $\mathcal{A}_{q,t}^{(s)}$ for easier reading. The operator $\mathcal{A}_{q,t}^{(s)}$ is understood as having a fixed value $t(x_s)$ of $t$ observed. Similarly we often write $\mathbb{E}_{q,t}$ to abbreviate $\mathbb{E}_{q,t=\underline{k}}$, with the same understanding.

**Proof of Lemma 3.1**  For convenience we repeat the statement of the lemma here.

**Lemma 3.1** *If $q_{\underline{k}}(x) > 0$ for all $\underline{k}$, then $\mathcal{A}_{q,t=\underline{k}}^{(s)}$ is a Stein operator for the conditional distribution of $X$ given $t(X) = \underline{k}$, and $\sum_s \mathcal{A}_{q,t=\underline{k}}^{(s)}$ is a Stein operator for the conditional distribution of $X$ given $t(X) = \underline{k}$.*

*Proof.* In order to show the assertion we prove that for $\mathbb{E}_{q,t=\underline{k}}$ denoting the conditional distribution of $X$ given $t(X) = \underline{k}$, the expectation $\mathbb{E}_{q,t=\underline{k}}[\mathcal{A}_{q,t}^{(s)} f]$ vanishes for all functions for which the expectation exists. Let $f$ be such a function. We have

$$\mathcal{A}_{q,t}^{(s)} f(x^{(s,1)}) = q(x^{(s,0)}|t(x_s) = \underline{k})(f(x^{(s,0)}) - f(x^{(s,1)}))$$
$$\mathcal{A}_{q,t}^{(s)} f(x^{(s,0)}) = q(x^{(s,1)}|t(x_s) = \underline{k})(f(x^{(s,1)}) - f(x^{(s,0)})).$$

Thus, $\mathbb{E}_{q,t}$,

$$\mathbb{E}_{q,t}[\mathcal{A}_{q,t}^{(s)} f] = \sum_{x_{-s}} \Big\{ \mathbb{1}(x_s = 1) p_t(x^{(s,1)}) q(y^{(s,0)}|t(y_s) = \underline{k})(f(x^{(s,0)}) - f(x^{(s,1)}))$$

$$-\mathbb{1}(x_s = 0) p_t(x^{(s,0)}) q(y^{(s,1)}|t(y_s) = \underline{k})(f(x^{(s,0)}) - f(x^{(s,1)})) \Big\}$$

$$= \sum_{x_{-s}} (f(x^{(s,0)}) - f(x^{(s,1)})) p_t(x^{(s,1)}) p_t(x^{(s,0)}) \{ \mathbb{1}(x_s = 1) - \mathbb{1}(x_s = 0) \}$$

$$= 0.$$

$\square$

**Proof of Theorem 3.2**  For convenience we repeat the theorem here.

**Theorem 3.2** *Assume that $\widehat{q}_t(x^{(s,1)})$ is a consistent estimator for $q_t(x^{(s,1)})$ as $L \to \infty$. Then for any function $f$ such that $\|\Delta f\| < \infty$ we have $\mathbb{E}_q[\mathcal{A}_{\widehat{q},t} f(x)] \to \mathbb{E}_q[\mathcal{A}_{q,t} f(x)] = 0$ as $L \to \infty$.*

*Proof.* We recall the notation that Equation (4). We have that

$$\mathcal{A}_{q,t} f(x) = \frac{1}{N} \sum_{s \in [N]} [q(x^{(s,1)}|t(x_{-s}) f(x^{(s,1)}) + q(x^{(s,0)}|t(x_{-s}) f(x^{(s,0)}) - f(x)]$$

so that

$$\mathcal{A}_{\widehat{q},t} f(x) - \mathcal{A}_{q,t} f(x) = \frac{1}{N} \sum_{s \in [N]} \{ (\widehat{q}(x^{(s,1)}|t(x_{-s})) - q(x^{(s,1)}|t(x_{-s})) f(x^{(s,1)})$$

$$+ (1 - \widehat{q}(x^{(s,1)}|t(x_{-s}) - (1 - q(x^{(s,1)}|t(x_{-s}))) f(x^{(s,0)}) \}$$

$$= \frac{1}{N} \sum_{s \in [N]} (\widehat{q}(x^{(s,1)}|t(x_{-s}) - q(x^{(s,1)}|t(x_{-s})) \Delta_s f(x).$$

Hence

$$|\mathcal{A}_{\widehat{q},t} f(x) - \mathcal{A}_{q,t} f(x)| \leq \|\Delta f\| \frac{1}{N} \sum_{s \in [N]} |\widehat{q}(x^{(s,1)}|t(x_{-s})) - q(x^{(s,1)}|t(x_{-s}))|.$$

Thus, if for all $s \in [N]$, as $L \to \infty$ we have $\widehat{q}(x^{(s,1)}|t(x_{-s})) - q(x^{(s,1)}|t(x_{-s})) \to 0$ for all $s$ then so does $|\mathcal{A}_{\widehat{q},t} f(x) - \mathcal{A}_{q,t} f(x)|$. The assertion follows from the assumption that $\widehat{q}_t(x^{(s,1)})$ is a consistent estimator for $q_t(x^{(s,1)})$ as $L \to \infty$. $\square$

**Proof of Theorem 3.3** For convenience we repeat the statement of the theorem here. Recall that a random graph model is *edge-exchangeable* if its edge indicator variables are finitely exchangeable. Often we just write *edge-exchangeable graph*. An ERGM is an example of an edge-exchangable graph.

**Theorem 3.3** *If the graph is edge-exchangeable, then* $\mathrm{AgraSSt}^2(\widehat{q}, t; x)$ *is a consistent estimator of*

$$\mathrm{gKSS}^2(q; x) = N^{-2} \sum_{s,s' \in [N]} \left\langle \mathcal{A}_q^{(s)} K(x, \cdot), \mathcal{A}_q^{(s')} K(\cdot, x) \right\rangle_{\mathcal{H}}.$$

For easier tractability the proof is organised in two steps.

1. First, Proposition A.1 shows that in an edge-exchangeable random graph model, $g_{\underline{k}}$ given in Equation (8) is a consistent estimator for $q(x^{(s,1)}|t(x_{-s}))$ as $NL \to \infty$.

2. Theorem A.2 uses these results to obtain a concentration bound for $\mathcal{A}_{\widehat{q}}$ from which then Theorem 3.3 follows.

Moreover theoretical guarantees for fixed $L$ which depend on the model are given. As the graph generator can generate as large a number $L$ of graphs as desired, these theoretical results can be used to determine $L$ which result in theoretical guarantees on deviations from the mean.

**Proposition A.1.** *Suppose that $X_1, \ldots, X_L$ are i.i.d. copies of the adjacency matrix of an edge-exchangeable random graph model. Let $s = (i, j)$ be a fixed vertex-pair. For $l = 1, \ldots, L$ and for a graph $X_l$ let $t_l^{(s)}$ denote a possibly multivariate statistic which is evaluated on the collection of indicator variables in $X_l$ except $X_{s,l}$. For a possible $t_l^{(s)}$ outcome $\underline{k}$, let $p(\underline{k}) = \mathbb{P}(t_l^{(s)} = \underline{k})$ and let $\underline{k}$ be such that $p(\underline{k}) \neq 0$. Set*

$$p(1; \underline{k}) = \mathbb{P}(X_s = 1 | t^{(s)} = \underline{k});$$

*let*

$$n(\underline{k}, s) = \sum_{l=1}^{L} X_{s,l} \mathbb{1}(t_l^{(s)} = \underline{k}) \quad and \quad n(\underline{k}^{(s)}) = \sum_{l=1}^{L} \mathbb{1}(t_l^{(s)} = \underline{k});$$

$$n(\underline{k}) = \sum_{s \in [N]} n(\underline{k}, s) \quad and \quad N_{\underline{k}} = \sum_{s \in [N]} n(\underline{k}^{(s)});$$

*and set*

$$g(\underline{k}) = \frac{n(\underline{k})}{N_{\underline{k}}} \mathbb{1}(N_{\underline{k}} \geq 1).$$

*Then $g(\underline{k}) \to p$ in probability as $NL \to \infty$. In particular, for all $\epsilon > 0$,*

$$\mathbb{P}\left[|\hat{g}(\underline{k}) - p(1; \underline{k})| > \epsilon\right] \leq \frac{4}{\epsilon^2 NL} \{p(s, \underline{k})[1 - p(s, \underline{k})p(\underline{k})] + 1 - p(\underline{k})\}.$$

*Proof.* Due to the exchangeability of the edges we have, with $s$ denoting a generic edge,

$$\mathbb{E}(n(\underline{k})) = N\mathbb{E}(n(\underline{k}, s)) = NLp(1; \underline{k})p(\underline{k}), \qquad Var(n(\underline{k})) = NLp(1; \underline{k})p(\underline{k})(1 - p(1; \underline{k})p(\underline{k})),$$

as well as

$$\mathbb{E}(N_{\underline{k}}) = N\mathbb{E}(n(\underline{k}^{(s)})) = NLp(\underline{k}), \qquad Var(N_{\underline{k}}) = NLp(\underline{k})(1 - p(\underline{k})).$$

To show convergence in probability, let $\epsilon > 0$. Then

$$\mathbb{P}\left[|\hat{g}(\underline{k}) - p(1, \underline{k})| > \epsilon\right] \leq \mathbb{P}\left[\left|\frac{n(\underline{k})}{\mathbb{E}(n(\underline{k}))}\mathbb{1}[n(\underline{k}) \geq 1] - p(1, \underline{k})\right| > \frac{1}{2}\epsilon\right]$$

$$+ \mathbb{P}\left[\left|n(\underline{k})\mathbb{1}[n(\underline{k}) \geq 1]\left(\frac{1}{n(\underline{k})} - \frac{1}{\mathbb{E}(n(\underline{k}))}\right)\right| > \frac{1}{2}\epsilon\right].$$

Note that $n(\underline{k})\mathbb{1}[n(\underline{k}) \geq 1] = n(\underline{k})$. By Chebychev's inequality,

$$\mathbb{P}\left[\left|\frac{n(\underline{k})}{\mathbb{E}(N_{\underline{k}})}\mathbb{1}[N_{\underline{k}} \geq 1] - p(1;\underline{k})\right| > \frac{1}{2}\epsilon\right] \leq \frac{4}{\epsilon^2 N^2 L^2 p(\underline{k})^2} NLp(1;\underline{k})p(\underline{k})(1 - p(1;\underline{k})p(\underline{k}))$$

$$= \frac{4}{\epsilon^2 NLp(\underline{k})}p(1;\underline{k})(1 - p(1;\underline{k})p(\underline{k}))$$

and

$$\mathbb{P}\left[\left|n(\underline{k})\mathbb{1}[N_{\underline{k}} \geq 1]\left(\frac{1}{n(\underline{k})} - \frac{1}{\mathbb{E}(N_{\underline{k}})}\right)\right| > \frac{1}{2}\epsilon\right] \leq \mathbb{P}\left[\frac{1}{\mathbb{E}(N_{\underline{k}})}\left|\mathbb{E}(N_{\underline{k}}) - N_{\underline{k}}\right| > \frac{1}{2}\epsilon\right]$$

$$\leq \frac{4}{\epsilon^2 NLp(\underline{k})}(1 - p(\underline{k})).$$

Summing the contributions completes the proof. $\qquad\square$

Proposition A.1 shows that in edge-exchangeable graphs, $\hat{g}(\underline{k})$ consistently estimates $q(x^{(s,1)}) = q(x^{(s,1)}|t(x_{-s}) = \underline{k})$. In an expanded version of Theorem 3.3 we show that the approximate Stein operator from Eq.(7),

$$\mathcal{A}_{\hat{q},t}f(x) := \frac{1}{N}\sum_{s\in[N]}\mathcal{A}_{\hat{q}(x^{(s)}|t(x_{-s}))}f(x),$$

with

$$\widehat{q}(x^{(s,1)}|t(x_{-s})) = g(x_{-s}), \quad \widehat{q}(x^{(s,0)}|t(x_{-s})) = 1 - g(x_{-s})$$

is a consistent estimator of

$$\mathcal{A}_{q,t}f(x) := \frac{1}{N}\sum_{s\in[N]}\mathcal{A}_{q(x^{(s)}|t(x_{-s}))}f(x).$$

We recall

$$\text{AgraSSt}^2(\hat{q},t,x) = \frac{1}{N^2}\sum_{s,s'\in[N]}h_x(s,s')$$

with

$$h_x(s,s') = \left\langle \mathcal{A}_{\hat{q},t}^{(s)}K(x,\cdot), \mathcal{A}_{\hat{q},t}^{(s')}K(\cdot,x)\right\rangle_{\mathcal{H}}.$$

We state the expanded version of Theorem 3.3 here.

**Theorem A.2.** *If the graph is edge-exchangeable then for any test function $f$ for which the Stein operator $\mathcal{A}_{q,t}f$ is well defined, and for all $\epsilon > 0$*

$$\mathbb{P}(|\mathcal{A}_{\hat{q},t}f(X) - \mathcal{A}_{q,t}f(X)| > \epsilon) \leq \frac{4}{\epsilon^2 NL(||\Delta f||)^{-2}}\left(\{p(1,\underline{k})[1 - p(1,\underline{k})p(\underline{k})] + 1 - p(\underline{k})\}\right).$$

*Moreover,* $\text{AgraSSt}^2(\hat{q},t,x)$ *is a consistent estimator of*

$$\text{gKSS}(x) = \frac{1}{N^2}\sum_{s,s'\in[N]}\left\langle \mathcal{A}_{q,t}^{(s)}K(x,\cdot), \mathcal{A}_{q,t}^{(s')}K(\cdot,x)\right\rangle_{\mathcal{H}}.$$

*Proof.* We have that

$$\mathcal{A}_{q,t}f(x) := \frac{1}{N}\sum_{s\in[N]}\mathcal{A}_{q(x^{(s)}|t(x_{-s}))}f(x).$$

and

$$\mathcal{A}_{\hat{q},t}f(x) := \frac{1}{N}\sum_{s\in[N]}\mathcal{A}_{\widehat{q}(x^{(s)}|t(x_{-s}))}f(x),$$

with

$$\widehat{q}(x^{(s,1)}|t(x_{-s})) = g(x_{-s}), \quad \widehat{q}(x^{(s,0)}|t(x_{-s})) = 1 - g(x_{-s})$$

so that

$$\mathcal{A}_{\widehat{q},t}f(x) - \mathcal{A}_{q,t}f(x) = \frac{1}{N}\sum_{s\in[N]}\{(g(\underline{k}) - q(x^{(s)}|t(x_{-s})))f(x^{(s,1)})$$
$$+ (1 - g(\underline{k}) - (1 - q(x^{(s)}|t(x_{-s}))))f(x^{(s,0)})\}$$
$$= \frac{1}{N}\sum_{s\in[N]}(g(\underline{k}) - q(x^{(s)}|t(x_{-s})))\{f(x^{(s,1)}) - f(x^{(s,0)})\}$$
$$= \frac{1}{N}\sum_{s\in[N]}(g(\underline{k}) - q(x^{(s)}|t(x_{-s})))\Delta_s f(x).$$

Hence

$$|\mathcal{A}_{\widehat{q},t}f(x) - \mathcal{A}_{q,t}f(x)| \leq ||\Delta f||\frac{1}{N}\sum_{s\in[N]}|g(\underline{k}) - q(x^{(s)}|t(x_{-s}))|.$$

With Proposition A.1 and using the edge-exchangeability,

$$\mathbb{P}(|\mathcal{A}_{\widehat{q},t}f(X) - \mathcal{A}_{q,,t}f(X)| > \epsilon) \leq \frac{4}{\epsilon^2 NL(||\Delta f||)^{-2}}\left(\{p(1,\underline{k})[1 - p(1,\underline{k})p(\underline{k})] + 1 - p(\underline{k})\}\right).$$

The fact that taking the sup over functions in the Hilbert space $\mathcal{H}$ does not spoil the convergence follows from the closed form representation of the sup of AgraSSt$^2$, see for example Equation (11) in [Xu and Reinert, 2021]. We have that

$$\mathrm{AgraSSt}^2(\hat{q}, t, x) = \frac{1}{N^2}\sum_{s,s'\in[N]}h_x(s,s')$$

where

$$h_x(s,s') = \left\langle \mathcal{A}_{\hat{q},t}^{(s)}K(x,\cdot), \mathcal{A}_{\hat{q},t}^{(s')}K(\cdot,x)\right\rangle_{\mathcal{H}}.$$

Hence,

$$\mathrm{AgraSSt}^2(\widehat{q}) - \mathrm{AgraSSt}^2(q)$$
$$= \frac{1}{N^2}\sum_{s,s'\in[N]}\left\langle \mathcal{A}_{\widehat{q},t}^{(s)}K(x,\cdot) - \mathcal{A}_{q,t}^{(s)}K(x,\cdot), \mathcal{A}_{\widehat{q},t}^{(s')}K(\cdot,x) - \mathcal{A}_{q,t}^{(s)}K(x,\cdot)\right\rangle_{\mathcal{H}}$$

and the first part gives the desired convergence as $L\to\infty$. $\qquad\square$

## A.2 Gaussian approximation for AgraSSt in ERGMs

As ERGMs are edge-exchangeable models, Theorem A.2 shows that the AgraSSt operator is a consistent estimator for the ERGM Glauber Stein operator. If the observed graph $x$ is a realisation of an ERGM then results from Xu and Reinert [2021] can be leveraged to obtain finer theoretical results.

First we detail the scaling for exponential random graph models which is used in the theoretical results which follow. For a graph $H$ on at most $n$ vertices $V(H)$ denote the vertex set, and for $x \in \{0,1\}^N$, denote by $t(H,x)$ the number of *edge-preserving* injections from $V(H)$ to $V(x)$; an injection $\sigma$ preserves edges if for all edges $vw$ of $H$ with $\sigma(v) < \sigma(w)$, $x_{\sigma(v)\sigma(w)} = 1$. For $v_H = |V(H)| \geq 3$ set

$$t_H(x) = \frac{t(H,x)}{n(n-1)\cdots(n-v_H+3)}.$$

If $H = H_1$ is a single edge, then $t_H(x)$ is twice the number of edges of $x$. In the exponent this scaling of counts matches Definition 1 in Bhamidi et al. [2011] and Sections 3 and 4 of Chatterjee and Diaconis [2013]. An ERGM for the collection $x \in \{0,1\}^N$ can be defined as follows.

**Definition A.3** ( Definition 1.5 in Reinert and Ross [2019]). Fix $n \in \mathbb{N}$ and $k \in \mathbb{N}$. Let $H_1$ be a single edge and for $l = 2, \ldots, k$ let $H_l$ be a connected graph on at most $n$ vertices; set $t_l(x) = t_{H_l}(x)$. For

$\beta = (\beta_1, \ldots, \beta_k)^\top \in \mathbb{R}^k$ and $t(x) = (t_1(x), \ldots, t_k(x))^\top \in \mathbb{R}^k$ $X \in \mathcal{G}_n^{lab}$ follows the exponential random graph model $X \sim \text{ERGM}(\beta, t)$ if for $\forall x \in \mathcal{G}_n^{lab}$,

$$q(X = x) = \frac{1}{\kappa_n(\beta)} \exp\left(\sum_{l=1}^{k} \beta_l t_l(x)\right).$$

531    Here $\kappa_n(\beta)$ is a normalisation constant.

In particular, under suitable conditions, the ERGM Glauber Stein operator is close to the $G(n, p)$ Stein operator. This result is already shown in Reinert and Ross [2019], Theorem 1.7, with details provided in the proof of Theorem 1 in Xu and Reinert [2021]. To give the result, a technical assumption is required, which originates in Chatterjee and Diaconis [2013], and is required in Reinert and Ross [2019]. For $a \in [0, 1]$, define the following functions [Bhamidi et al., 2011, Eldan and Gross, 2018], with the notation in Definition A.3 for $\text{ERGM}(\beta, t)$:

$$\Phi(a) := \sum_{l=1}^{k} \beta_l e_l a^{e_l - 1}, \quad \varphi(a) := \frac{1 + \tanh(\Phi(a))}{2}$$

532    where $e_l$ is the number of edges in $H_l$.

533    *Assumption 1.* (1) $\frac{1}{2}|\Phi|'(1) < 1$. (2) $\exists a^* \in [0, 1]$ that solves the equation $\varphi(a^*) = a^*$.

534    The value $a^*$ will be the edge probability in the approximating Bernoulli random graph, $\text{ER}(a^*)$.
535    The following result holds.

**Proposition A.4.** *Let $q(x) = \text{ERGM}(\beta, t)$ satisfy Assumption 1 and let $\tilde{q}$ denote the distribution of $ER(a^*)$. Then there is an explicit constant $C = C(\beta, t, K)$ such that for all $\epsilon > 0$,*

$$\frac{1}{N} \sum_{s \in N} \mathbb{E}|(\mathcal{A}_q^{(s)} f(Y) - \mathcal{A}_{\tilde{q}}^{(s)} f(Y))| \leq ||\Delta f|| \binom{n}{2} \frac{C(\beta, t)}{\sqrt{n}}.$$

536    *Moreover, for $f \in \mathcal{H}$ equipped with kernel $K$, let $f_x^*(\cdot) = \frac{(\mathcal{A}_q - \mathcal{A}_{\tilde{q}})K(x, \cdot)}{||(\mathcal{A}_q - \mathcal{A}_{\tilde{q}})K(x, \cdot)||_{\mathcal{H}}}$. Then there is an*
537    *explicit constant $C = C(\beta, t, K)$ such that for all $\epsilon > 0$,*

$$\mathbb{P}(|\text{gKSS}(q, X) - \text{gKSS}(\tilde{q}, Y)| > \epsilon) \tag{12}$$

$$\leq \left\{ ||\Delta(\text{gKSS}(q, \cdot))^2||(1 + ||\Delta\text{gKSS}(q, \cdot)||) + 4 \sup_x(||\Delta f_x^*||^2) \right\} \binom{n}{2} \frac{C}{\epsilon^2 \sqrt{n}}. \tag{13}$$

538    *Proof.* The assertion follows immediately from the proof of Theorem 1 in Xu and Reinert [2021]. $\square$

539    The approximation with a Bernoulli random graph is useful as for a Bernoulli random graphs a
540    normal approximation for its gKSS is available in Xu and Reinert [2021], under suitable assumptions.

541    *Assumption 2.* Let $\mathcal{H}$ be the RKHS associated with the kernel $K : \{0, 1\}^N \times \{0, 1\}^N \to \mathbb{R}$ and for
542    $s \in [N]$ let $\mathcal{H}_s$ be the RKHS associated with the kernel $l_s : \{0, 1\} \times \{0, 1\} \to \mathbb{R}$. Then

543    i) $\mathcal{H}$ is a tensor product RKHS, $\mathcal{H} = \otimes_{s \in [n]} \mathcal{H}_s$;
544    ii) $k$ is a product kernel, $k(x, y) = \otimes_{s \in [N]} l_s(x_s, y_s)$;
545    iii) $\langle l_s(x_s, \cdot), l_s(x_s, \cdot) \rangle_{\mathcal{H}_s} = 1$;
546    iv) $l_s(1, \cdot) - l_s(0, \cdot) \neq 0$ for all $s \in [N]$.

547    These assumptions are satisfied for example for the suitably standardised Gaussian kernel $K(x, y) =$
548    $\exp\{-\frac{1}{\sigma^2} \sum_{s \in [N]} (x_s - y_s)^2\}$.

549    Letting $|| \cdot ||_1$ denote $L_1$-distance, and $\mathcal{L}$ denote the law of a random variable, Xu and Reinert [2021]
550    show the following normal approximation.

551    **Theorem A.5** (Theorem 2 in Xu and Reinert [2021]). *Let $Y$ have the distribution $\tilde{q}$ of a Bernoulli*
552    *random graph $ER(a^*)$ as in Proposition A.4. Assume that the conditions i) - iv) in Assumption 2*
553    *hold. Let $\mu = \mathbb{E}[\text{gKSS}^2(\tilde{q}, Y)]$ and $\sigma^2 = Var[\text{gKSS}^2(\tilde{q}, Y)]$. Set $W = \frac{1}{\sigma}(\text{gKSS}^2(\tilde{q}, Y) - \mu)$ and*

let $Z$ denote a standard normal variable, Then there is an explicit constant $C = C(a^*, l_s, s \in [N])$ such that

$$||\mathcal{L}(W) - \mathcal{L}(Z)||_1 \leq \frac{C}{\sqrt{N}}.$$

Thus a normal approximation for the approximating gKSS can then be used to assess the theoretical behaviour of AgraSSt as follows.

**Corollary A.6.** *Let the assumptions Proposition A.4 and Theorem A.5 be satisfied. With the notation of Theorem A.5, assume that the RKHS kernel $K$ is such that the right hand side of Proposition A.4 is $o(n)$. Then $\frac{1}{\sigma}(\mathrm{AgraSSt}(\widetilde{q}(x^{(s)}|t(x_{-s}))) - \mu)$ is approximately standard normally distributed as $N \to \infty$.*

*Proof.* For all $\epsilon > 0$,

$$\mathbb{P}\left[\left|\mathrm{AgraSSt}(\widetilde{q}(x^{(s)}|t(x_{-s}))) - \mathrm{gKSS}(a^*)\right| > \epsilon\right]$$
$$\leq \quad \mathbb{P}\left[\left|\mathrm{AgraSSt}(\widetilde{q}(x^{(s)}|t(x_{-s}))) - \mathrm{gKSS}(q)\right| > \frac{1}{2}\epsilon\right] + \mathbb{P}\left[|\mathrm{gKSS}(q) - \mathrm{gKSS}(a^*)| > \frac{1}{2}\epsilon\right].$$

The first summand tends to 0 as $N \to \infty$ due to Theorem 3.2 and the second summand tends to 0 due to Proposition A.4. That $\mathrm{gKSS}(a^*)$ is approximately normally distributed with the appropriate scaling follows from Theorem A.5. $\qquad\square$

The theoretical behaviour of the subsampling version $\widehat{\mathrm{AgraSSt}}(\widetilde{q}(x^{(s)}|t(x_{-s})))$ is addressed in Proposition 3.4. A detailed examination of the choice of kernel $K$ such that the assumptions of Corollary A.6 are satisfied is left for future work.

# B  Additional background

In this section, we present additional background to complement the discussions in the main text.

## B.1   Parameter estimation for random graphs

Estimating parameters for parametric models is possible *only* when the parametric family is *explicitly specified*. For instance, in the synthetic example for E2ST model shown in Section 5.1, $\hat{\beta}_l$ can be estimated for $\beta_l$ since the edge, 2Star and triangle statistics are specified. There are various approaches for parameter estimation.

**Maximum likelihood**   Maximum likelihood is a popular approach for parameter estimation in random graph models. A complication arises because its probability mass function from Eq.(1),

$$q(X = x) = \frac{1}{\kappa_n(\beta)} \exp\left(\sum_{l=1}^{k} \beta_l t_l(x)\right).$$

involves a normalisation constant $\kappa_n(\beta) = \sum_x \exp\{\sum_{l=1}^{k} \beta_l t_l(x)\}$ which is generally intractable and needs to be estimated for performing MLE. For this task, Markov chain Monte-Carlo maximum likelihood estimation (MCMCMLE) for ERGM has been developed by Snijders [2002]. When the network size is large, accurate estimation for the normalised $\kappa_n(\beta)$ requires large amount of Monte-Carlo samples and is hence computationally expensive.

**Maximum pseudo-likelihood estimator**   To alleviate the problem associated with the normalising constant, Maximum Pseudo-likelihood Estimation (MPLE) [Besag, 1975] has been developed for ERGMs, see Strauss and Ikeda [1990] and also Schmid and Desmarais [2017]. MPLE factorises the conditional edge probability to approximate the exact likelihood,

$$q(x) = \Pi_{s \in [N]} q(x^s | x_{-s}). \tag{14}$$

587 For ERGMs the conditional distribution $q(x^s|x_{-s})$ does not involve the normalising constant and
588 can hence be computed more efficiently than the MLE. However, in general the MPLS is not
589 consistent for ERGMs as the edges are generally non-independent. The consistency of MPLE for
590 Boltzmann machines is shown in Hyvärinen [2006]. A thorough comparison of MCMCMLE and
591 MPLE estimation in ERGMs can be found in Van Duijn et al. [2009].

592 **Contrastive divergence** Estimation based on contrastive divergence (CD) [Hinton, 2002] has also
593 been developed for ERGM estimation [Hunter and Handcock, 2006]. Contrastive divergence runs a
594 small number of Markov chains simultaneously for $T$ steps and estimates the gradient based on the
595 differences between initial values and values after $T$ steps in order to find a maximum. Convergence
596 results for exponential family models are shown in Jiang et al. [2018]. CD can provide a useful
597 balance between computationally expensive but accurate MCMCMLE and fast but inconsistent
598 MPLE.

## B.2 Kernel Stein discrepancies and kernel-based nonparametric hypothesis testing

600 The task of hypothesis testing involves the comparison of distributions $p$ and $q$ that are significantly
601 different with respect to the size of the test, denoted by $\alpha$. In nonparametric tests, the distributions
602 are not assumed to be in any parametric families and test statistics are often based on ranking of
603 observations. In contrast, parametric tests, such as a Student t-test or a normality test, assume a
604 pre-defined parametric family to be tested against and usually employ a particular summary statistics
605 such as means or standard deviations. Recent advances in nonparametric test procedures introduce
606 RKHS functions which can be rich enough to distinguish distributions whenever they differ. Below
607 we detail two instances which are relevant for the main paper.

608 We start with a terse review of **kernel Stein discrepancy (KSD) for continuous distributions**
609 developed to compare and test distributions [Gorham and Mackey, 2015, Ley et al., 2017]. Let $q$
610 be a smooth probability density on $\mathbb{R}^d$ that vanishes at the boundary. The operator $\mathcal{A}_q : (\mathbb{R}^d \to$
611 $\mathbb{R}^d) \to (\mathbb{R}^d \to \mathbb{R})$ is called a *Stein operator* if the following *Stein identity* holds: $\mathbb{E}_q[\mathcal{A}_q f] = 0$,
612 where $f : \mathbb{R}^d \to \mathbb{R}^d$ is any bounded smooth function. A suitable function class $\mathcal{F}$ is such that if
613 $\mathbb{E}_p[\mathcal{A}_q f] = 0$ for all functions $f \in \mathcal{F}$, then $p = q$ follows. It is convenient to take $\mathcal{F} = B_1(\mathcal{H})$, the
614 unit ball of a large enough RKHS with bounded kernel $K$. The kernel Stein discrepancy (KSD)
615 between two densities $p$ and $q$ based on $\mathcal{A}_q$ is defined as

$$\text{KSD}(p\|q, \mathcal{H}) = \sup_{f \in B_1(\mathcal{H})} \mathbb{E}_p[\mathcal{A}_q f]. \tag{15}$$

616 Under mild regularity conditions, for a particular choice of $\mathcal{A}$ called Langevin operator,
617 $\text{KSD}(p\|q, \mathcal{H}) \geq 0$ and $\text{KSD}(p\|q, \mathcal{H}) = 0$ if and only if $p = q$ [Chwialkowski et al., 2016], in
618 which case KSD is a proper discrepancy measure between probability densities.

619 The KSD in Eq.(15) can be used to test the model goodness-of-fit as follows. One can show that
620 $\text{KSD}^2(p\|q, \mathcal{H}) = \mathbb{E}_{x,\tilde{x} \sim p}[h_q(x, \tilde{x})]$, where $x$ and $\tilde{x}$ are independent random variables with density $p$
621 and $h_q(x, \tilde{x})$ is given in explicit form which does not involve $p$,

$$h_q(x, \tilde{x}) = \langle \mathcal{A}_q K(x, \cdot), \mathcal{A}_q K(\cdot, \tilde{x}) \rangle_{\mathcal{H}}. \tag{16}$$

622 Given a set of samples $\{x_1, \ldots, x_n\}$ from an unknown density $p$ on $\mathbb{R}^d$, to test whether $p = q$, the
623 statistic $\text{KSD}^2(p\|q, \mathcal{H})$ can be empirically estimated by independent samples from $p$ using a $U$- or $V$-
624 statistic. The critical value is determined by bootstrap based on weighted chi-square approximations
625 for $U$- or $V$-statistics. For goodness-of-fit tests of discrete distributions when i.i.d. samples are
626 available, a kernel discrete Stein discrepancy (KDSD) has been proposed in Yang et al. [2018].

**Goodness-of-fit Testing** aims to check the null hypothesis $\mathfrak{H}_0 : p = q$ against the general alternative
$\mathfrak{H}_1 : p \neq q$ when the target distribution $q$ is explicitly specified. Given sample(s) from the *unknown*
distribution $p$ and an explicit density $q$, $\mathfrak{H}_0$ is assessed using a chosen test statistic, usually a
discrepancy measure, $D(q\|p)$, between $p$ and $q$, which can be estimated empirically. Kernel-based
hypothesis tests on goodness-of-fit for continuous distributions $q$ use the kernel Stein discrepancy

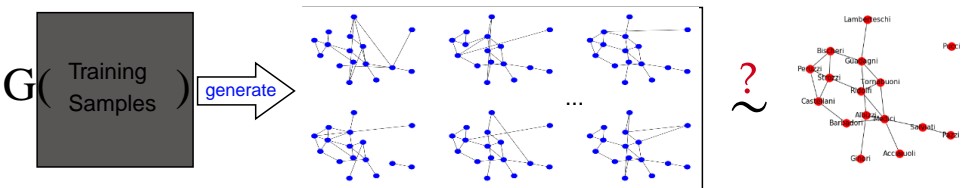

Figure 3: Assessing trained graph generators.

(KSD) in Section B.2 as the test statistic. Given samples $x_1, \ldots, x_n$ from the *unknown* density $p$, $\mathrm{KSD}^2(p\|q, \mathcal{H})$ in Eq.(15) is estimated via the $V$-statistic

$$\widehat{\mathrm{KSD}}^2(p\|q, \mathcal{H}) = \frac{1}{n^2} \sum_{i,j} h_q(x_i, x_j);$$

recall that $h_q(x_i, x_j) = \langle \mathcal{A}_q K(x_i, \cdot), \mathcal{A}_q K(x_j, \cdot) \rangle_{\mathcal{H}}$ from Eq.(16). The null distribution of this test statistic involves integral operators that are not available in close form; often it is simulated using a wild-bootstrap procedure [Chwialkowski et al., 2014]. With the (simulated) null distribution, the critical value of the test can be estimated to decide whether the null hypothesis is rejected at test level $\alpha$. In this way, a general method for nonparametric testing of goodness-of-fit on $\mathbb{R}^d$ is obtained, which is applicable even for models with an intractable normalising constant.

**Two-sample Testing** aims to determine whether two sets of samples are drawn from the same distribution, i.e. instead of $q$ being available in density form as in the goodness-of-fit setting, $q$ is only accessible through samples. Maximum mean embedding (MMD) test are often used for this two-sample problem [Gretton et al., 2007]. These tests are based on the kernel mean embedding of a distribution,

$$\mu_p := \mathbb{E}_{x \sim p}[k(x, \cdot)] = \int_{\mathcal{X}} k(x, \cdot) dp(x) \in \mathcal{H}, \tag{17}$$

whenever $\mu_p$ exist. Similar to KSD, MMD takes the supremum over unit ball RKHS functions;

$$\mathrm{MMD}(p\|q) = \sup_{f \in B_1(\mathcal{H})} \left| \mathbb{E}_p[f] - \mathbb{E}_q[f] \right| = \|\mu_p - \mu_Q\|_{\mathcal{H}}. \tag{18}$$

With samples $x_1, \ldots x_m \sim p$ and $y_1, \ldots, y_n \sim q$, MMD can be estimated empirically via $U$-statistics,

$$\widehat{\mathrm{MMD}}_u^2(p\|q) = \frac{1}{m(m-1)} \sum_{i \neq i'} k(x_i, x_{i'}) + \frac{1}{n(n-1)} \sum_{j \neq j'} k(y_j, y_{j'}) - \frac{2}{mn} \sum_{ij} k(x_i, y_j). \tag{19}$$

In such kernel-based two-sample tests, the null distribution can be obtained via a permutation procedure [Gretton et al., 2007]; this procedure can be more robust compared to a wild-bootstrap procedure, especially when the kernels need to be optimised [Gretton et al., 2012, Jitkrittum et al., 2016, Liu et al., 2020, 2021].

The two-sample procedure can also be applied to verify model assumptions when the model is not directly accessible through its distribution but through generated samples. Such a strategy has been considered as benchmark testing procedure in various studies for goodness-of-fit tests [Jitkrittum et al., 2017, Xu and Matsuda, 2020, 2021]. Despite lower test power compared to the corresponding state-of-the-art KSD-based tests and higher computational cost due to additional empirical estimation for the distribution $q$, the MMD-based tests are competitive with a simpler derivation in complicated testing scenarios [Xu and Matsuda, 2020, 2021], and they can outperform non-kernel based goodness-of-fit tests as discussed in Xu and Matsuda [2020].

## C  Visual illustrations of the assessment procedures

While AgraSSt is illustrated in Figure.1, we provide an additional visualisation emphasising different tasks for which AgraSSt can be applied. In Section 4.1 in the main text, we mentioned two features of our proposed AgraSSt procedure:

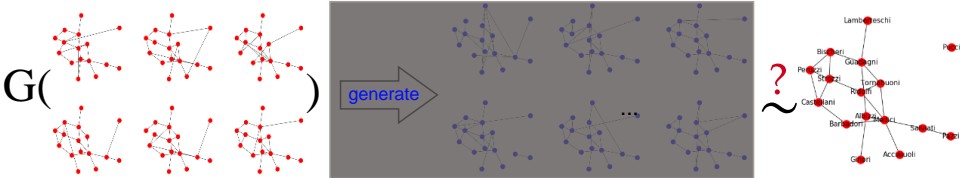

Figure 4: Criticising training quality for generative models.

1. Regardless of the learning or training procedures (masked in grey), AgraSSt can test a given generator $G$ that is only accessible through its generated samples as shown in Figure.3. In this setting, we do not need to know how the generator $G$ is obtained and the focus is the assessment of a particular generator $G$ itself.

2. Moreover, we are also interested in understanding the quality and capability of training procedures of (deep) generative models. As illustrated in Figure.4, a generator $G$ is trained from the same distribution as the input graph, e.g. ERGMs. The focus in this setting is to assess the training procedure of the generative model. (The samples generated are masked in grey.) For instance, for $G$ trained from the Florentine marriage network [Padgett and Ansell, 1993], we may like to understand whether the generative model can be trained to generate graphs that resemble the Florentine marriage network.

# D Additional experimental results and discussions

## D.1 Generating reliable samples

To illustrate how AgraSSt can be used to select sample batches, Figure.5 shows three sample batches of size 8 for the Karate club network of Zachary [1977], including the corresponding $p$-values for the displayed sample batches. Here we would expect to detect some community structure in the networks; only the sample batch from CELL captures this feature at least to some extent and has $p$-value which would not lead to rejection at the 5% level. This finding chimes with the results from Table 2; AgraSSt rejects both GraphRNN and NetGAN as synthetic data generators, but does not reject CELL.

## D.2 Additional case study: Padgett's Florentine network

Padgett's Florentine network [Padgett and Ansell, 1993]. has 16 vertices and 20 edges; in Xu and Reinert [2021] the hypothesis that it is an instance of a $G(n, p)$ model could not be rejected.

|  | AgraSSt | Deg | MDdeg | TV_deg |
|---|---|---|---|---|
| GraphRNN | 0.01 | 0.11 | 0.26 | 0.03 |
| NetGAN | 0.16 | 0.18 | 0.09 | 0.06 |
| CELL | 0.23 | 0.36 | 0.69 | 0.18 |

Table 3: $p$-values for models trained from Florentine marriage network; 100 samples to simulate the null; rejected null at significant level $\alpha = 0.05$ is marked red.

The $p$-values for different tests are shown in Table.3. The Florentine marriage network has edge density $q = 0.167$, while the trained CELL has $\widehat{q} = 0.165$ which is a close approximation. GraphRNN generates graphs with higher edge density $\widehat{q} = 0.188$. NetGAN generate samples with $\widehat{q} = 0.176$, not too different from the null, which is not rejected at $\alpha = 0.05$. This is different from what we see in the ERGM case above. This discrepancy may arise as the Florentine network is small with $n = 16$ and not highly clustered, with average local clustering coefficient $0.191$.

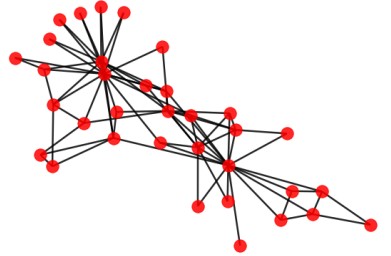

(a) The Karate Club network (vertices in red)

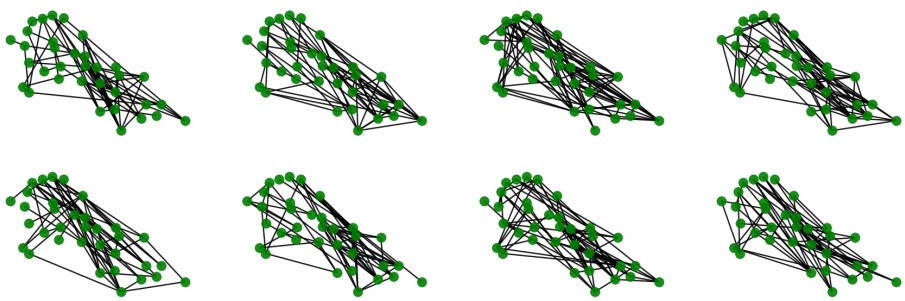

(b) Samples generated from GraphRNN model trained on Karate Club network (vertices in green)

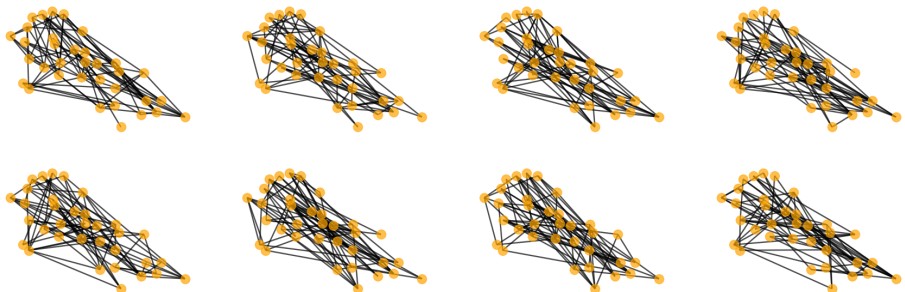

(c) Samples generated from NetGAN model trained on Karate Club network (vertices in orange)

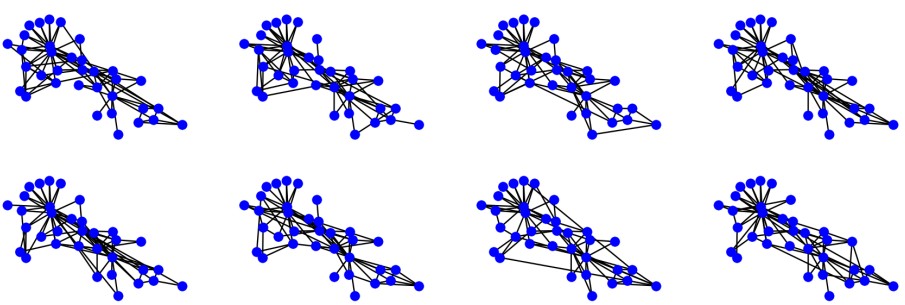

(d) Samples generated from CELL model trained on Karate Club network (vertices in blue) 0.26

Figure 5: The Karate Club network Zachary [1977] and three sample batches of size 8 from different graph generators. The $p$-value for GraphRNN samples in (b) is 0.00, for NetGAN samples in (c) the $p$-value is 0.01; for CELL samples in (d) the $p$-value is 0.26.

**Sample batch selection**    With CELL being deemed a good generator for the Florentine marriage network, we generate a sample batch of size 30 and check the sample quality. Most sample batches produce a $p$-value above $\alpha = 0.05$ until the 8th batch, which has $p$-value $0.03 < \alpha$. AgraSSt would recommend not taking this batch. A visual illustration is shown in Figure.6.

To investigate these batches, we note that the Florentine marriage network has 3 triangles, while the batch being rejected has a significantly lower average number of triangles, namely 1.2. Despite a well estimated edge density, this batch produces a low $p$-value. This batch, identified by AgraSSt as less reliable, may not be very suitable for downstream tasks and it may be better to generate another batch instead. In contrast, the batch with $p$-value 0.75 has 2.28 triangles on average while the batch with $p$-value 0.37 has 2.04 triangles on average; these averages are closer to the observed number of triangles.

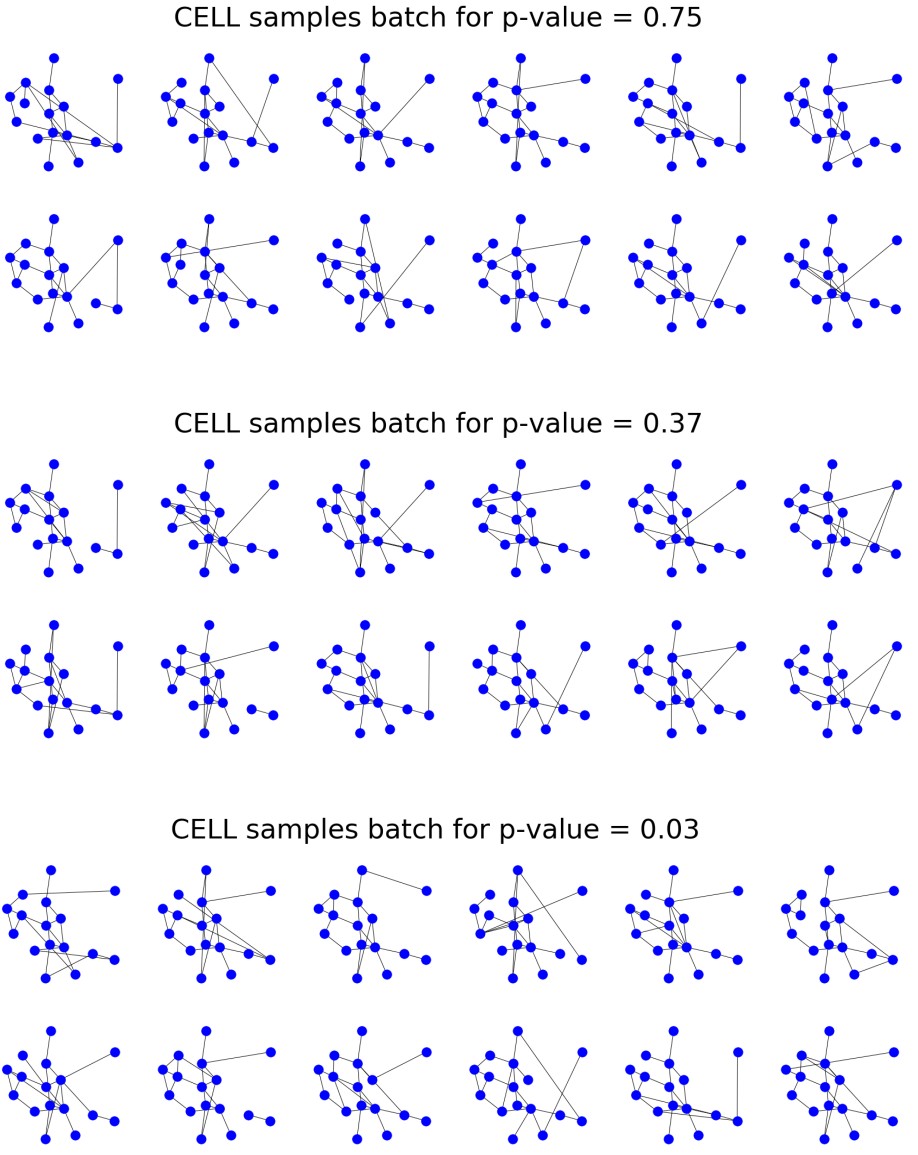

Figure 6: Samples from small size batches generated from CELL trained on the Florentine marriage network, with AgraSSt $p$-values. The first two batches would be deemed suitable by AgraSSt, while AgraSSt would not accept the third sample at the 5% significance level.

 **D.3 Experiments with other network statistics**

AgraSSt can incorporate any user-defined network statistics. Table 4 and Table 5 show additional results in the settings of Figure 2(b) and Table 1, respectively. As AgraSSt network statistics $t(x_{-s})$, we introduce D3, which is based on the multivariate statistics (edges((i,j)),deg(i),deg(j)), and we introduce Tri, which is based on the number of common neighbours of $i$ and $j$. The edge based AgraSSt from the main text is added in grey for comparison.

| perturbed $\beta_2$ | -0.60 | -0.40 | -0.20 | 0.00 | 0.20 |
|---|---|---|---|---|---|
| AgraSSt_D3 | 0.93 | 0.87 | 0.60 | 0.06 | 1.00 |
| AgraSSt_Tri | 0.82 | 0.71 | 0.35 | 0.07 | 1.00 |
| AgraSSt (main) | 0.95 | 0.89 | 0.68 | 0.04 | 1.00 |

Table 4: Rejection Rate for the setting in Figure 2(b).

| Models | GraphRNN | NetGAN | CELL | MC |
|---|---|---|---|---|
| AgraSSt_D3 | 0.31 | 0.66 | 0.10 | 0.03 |
| AgraSSt_Tri | 0.28 | 0.32 | 0.12 | 0.06 |
| AgraSSt (main) | 0.42 | 0.81 | 0.05 | 0.04 |

Table 5: Rejection Rate for the setting in Table 1.

In the Florentine network example, D3 has $p$-values 0.04 for GraphRNN, 0.11 for NetGAN, and 0.74 for CELL. Tri has $p$-values 0.02 for GraphRNN, 0.01 for NetGAN, and 0.12 for CELL. Overall, the results are mainly comparable to using AgraSSt based on the number of edges, although Tri rejects NetGAN for the Florentine marriage network, thus picking up on NetGAN struggling to reproduce local clustering.

**D.4 Additional discussions on distance-based test statistics**

A classical approach for goodness-of-fit testing in ERGMs is the graphical test by Hunter et al. [2008]. The idea is to simulate sample graphs under the null distribution statistics and create box plots of some relevant network statistics; add to these plots the network statistics in the observed network, as a solid line for comparison, which is illustrated in Figure.7. The box plot is used to check whether the observed network is "very different" from the simulated null samples. This graphical test procedure can be translated into Monte Carlo tests. It is natural to adapt such procedure to implicit models from which samples can be obtained. Figure.7 plots standard network statistics from Hunter et al. [2008] for samples from a fitted $G(n, p)$ generator (ER Approximate) and a learned GraphRNN generator of the Florentine marriage network described in more detail in Appendix D.2. The bold black line indicates the distribution of statistics for the Florentine marriage network.

The distribution of network statistics is then quantified via Total Variation (TV) distance [Xu and Reinert, 2021], based on which goodness-of-fit testing with $p$-values can be conducted. We find that while the fitted ER generator shows a reasonable fit for all summary statistics, the GraphRNN generator does not match the Florentine marriage network very well for dyad-wise shared partners and the triad census.

**D.5 Efficiency results**

Table.6 presents the computational runtime (RT) and the test construction time (CT) for AgraSSt and its comparison methods from Section 5.1.1 with the simulation setup as in Section 5.1.2. As a measure of accuracy, the variance (Var) of the simulated (or estimated) test statistics under the null distribution is also included.

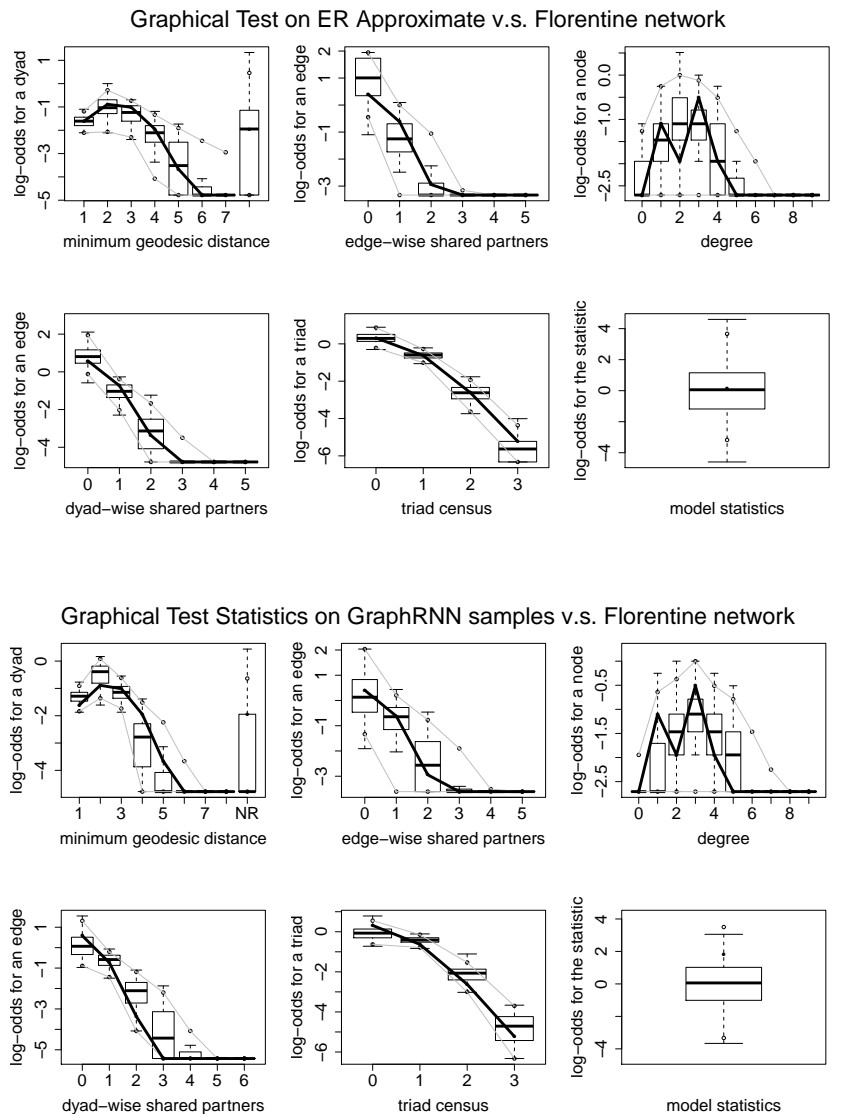

Figure 7: Graphical test illustrations on samples from generators learned on the Florentine marriage network

The parameter estimation in **Param** depends on a computationally efficient method which is based on MPLE [Schmid and Desmarais, 2017] in Eq.(14). **AgraSSt** takes longer to compute mainly due to the computation of graph kernels, e.g. Weisfeiler-Lehman kernel [Shervashidze et al., 2011]. We note that for implicit models, the estimation step in AgraSSt relies on generating samples from the model so that the the computational advantage[4] of the Stein based test over graphical goodness-of-fit tests[5] reduces compared to gKSS. **MDdeg** is computationally expensive due to the estimation of an inverse covariance matrix. While providing fast computation and estimation, **Deg** and **Param** sacrifice test power through a large variance of the test statistics. Estimating the full degree distribution, the total variation distance method **TV_deg**, based only on degrees, is competitive with AgraSSt; we recall that in our simulation results from Section 5.1.2 **TV_deg** was less powerful than AgraSSt. Here **MDdeg** is outperformed by the other test statistics.

|       | AgraSSt | Deg    | Param | MDdeg   | TV_deg |
|-------|---------|--------|-------|---------|--------|
| RT(s) | 0.141   | 0.0006 | 0.014 | 0.831   | 0.002  |
| CT(s) | 28.656  | 0.277  | 2.963 | 162.912 | 0.555  |
| Var   | 0.23    | 8.38   | 1.43  | 15.84   | 0.28   |

Table 6: Computational efficiencies and uncertainty in estimates. RT: runtime for one test; CT: construction time for the test class, including generating 500 samples for relevant estimation and 200 samples for simulating from the null distribution; Var: the estimated variance under the simulated null distribution. Both RT and CT are in seconds.

### D.6 Additional implementation details

For GraphRNN, we use batch size 128, epoch 1000 for training, 100 for testing, and learning rate 0.003. For CELL, we use learning rate 0.01, and weight decay 1e-7. For NetGAN, we use batch size 128, epoch 50, generator size and discriminator size both 128, and learning rate 0.0003.

We note that training NetGAN [Bojchevski et al., 2018] with the Florentine and with the Karate Club network may encounter some generator instability and hence early stopping can be useful. Without early stopping, the training loss for the generator increases during training, although it should be decreasing. Figure.8 shows the training loss on the generator in NetGAN as well as on the critic (or discriminator) in NetGAN. Figure.8 plots the loss every 200 training epochs. We can see from Figure.8(a) that the generator loss starts to be unstable and then increases after 50 points, i.e. 10,000 epochs. Hence we use only 10,000 epochs for training.

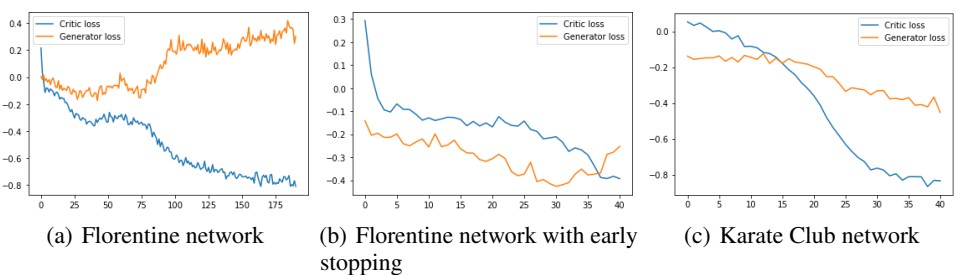

(a) Florentine network     (b) Florentine network with early stopping     (c) Karate Club network

Figure 8: Training loss (y-axis) for NetGAN [Bojchevski et al., 2018]; plotted against every 200 training epochs (x-axis).

---

[4]These results on gKSS are shown in Supplementary Material D in Xu and Reinert [2021].

[5]The graphical test [Hunter et al., 2008] is computed based on generating a large amount of samples from the null distribution.