# OpenReview forum: "AgraSSt: Approximate Graph Stein Statistics for Interpretable Assessment of Implicit Graph Generators"
_NeurIPS.cc/2022/Conference — NeurIPS 2022 Accept_

### Official Review · Reviewer_QFs7 · 2022-06-28

**Rating:** 6
**Confidence:** 3
**Soundness:** 3 good
**Presentation:** 2 fair
**Contribution:** 3 good

**Summary:**

The authors propose AgraSSt, a method that can be used to compare graphs generated by (e.g., deep) models to the single reference graph that the model was trained on. The method evaluates a chosen set of graph statistics (e.g., edge density, triangle count) on the generated graphs and compares their statistics with those on the reference graph. The method can reject the null hypothesis that the generated graphs come from the same distribution as the reference graph.

**Questions:**

* Eq. (4), what is $t(x_s)$? Should this be $t(x_{-s})$?
* Are there any constraints on what are valid graph statistics to include in AgraSSt?
* In Section 5.2.3, which set of graph statistics was used for the karate club experiments?

> GraphRNN and NetGAN samples exhibit a single large component rather than two fairly separated communities in the Karate Club network. This difference is picked up by AgraSSt, Deg and TV_deg [...]

I wonder how this difference between a single large component and two mostly separated community is picked up by the models. The statistics in the text (edge count, triangle count, degree statistics, ...) do not seem to be sensitive to such a high-level feature of the graph.

* Relatedly, (how) do different choices of graph statistics affect AgraSSt's ability to distinguish graphs from a different distribution than the reference graph which are specifically tailored to reproduce specific statistics? For instance, the authors mention that NetGAN produces graphs with similar densities but very different triangle counts compared to the reference graph. How can we be confident that we have not missed important graph statistics?

* The conclusion mentions Appendix E, but there are only appendices up to the letter D in the supplementary material.

**Limitations:**

* The authors do not discuss limitations of their method. The most important (potential) limitations I have in mind are:
  * What is the complexity of the approach? What is the typical runtime in practice, depending on the graph size?
  * To rephrase my point from the "questions" section: it appears that the network statistics used by the authors are mostly simple or "low-level" (e.g., number of triangles, edge density). What kind of graphs/ samples cannot be distinguished by the test used in the paper?

**Strengths And Weaknesses:**

Strengths:
* The approach proposed by the authors appears to be theoretically well justified, sound, and novel.
* The approach is general-purpose and is not limited to certain types of graph generators.
* The authors study recent deep-learning-based graph generators.

Weaknesses:
* It is not sufficiently clear which graph statistics are used in AgraSSt for which experiments; relatedly, it is not clear how the test could pick up high-level graph features such as power law exponent or the existence of communities.
* The method is only run on synthetic and very small toy-like real-world graphs. Possibly relatedly, there is no mentioning of the computational complexity of their method (or runtimes). This means that the real-world applicability and thus significance of the approach is not clear.
* The method description is quite dense and suffers from mathiness, making it hard to follow at times.

---

> ### Author Response · Authors · 2022-08-01
> **Advantages of AgraSSt as model assessment tools, computational complexity, and several clarifications**
>
> Thank you for your assessment.
>
> The graph statistics used in the experiments are detailed in Section 5.1.2. As noted in that section, ''Edges'' performs best, and hence it was used in real-world application. This is now made clearer in the text. In addition, in SI D.3 we explore other network statistics.   What you consider a weakness, namely which graph statistics to use, we view as a strength as it adds flexibility. Guidance can be taken from social network analysis and from graph limit theory; they both consider subgraph counts as summary statistics. AgraSSt is designed to assess synthetic data generators, not to pick up particular features of the observed network. Once a synthetic data generator has passed the assessment, one can generate samples from it and use these to study features of the original network, in a downstream analysis which is not carried out in the paper. In the Karate network example, Fig.5 in SI D, if the interest lies in community detection, one could calculate the modularity of the accepted and rejected samples and compare them to the modularity of the Karate network. This analysis would point to the modularity being an important feature of the network. Trying this out the modularity for the Karate club and for the average based on 8 samples each from the graph generators is
> given in the following table.
>
> |    Karate Club Data    | GraphRNN | NetGAN | CELL | Original network |
> | ----------- | ----------- | ----------- |----------- |----------- |
> | mean modularity ($\pm$std) | 0.4104 ($\pm$0.0357) | 0.3628 ($\pm$0.0479) | 0.3668 ($\pm$0.0177) | 0.3806  |
>
>
> The CELL generator, which performed best in AgraSSt, has the modularity which is closest to the observed modularity. Moreover, it has by far the smallest standard deviation.
>
> Similarly, if the interest is in estimating a power law exponent, accepted and rejected samples from an accepted synthetic data generator can be employed. We emphasise that such downstream analysis is not a focus of AgraSSt; rather, AgraSSt enables such a downstream analysis.
>
> You notice as a weakness that there is no mention of the computational complexity or the runtime. The runtimes are indeed reported in Fig.7 of SI D.5. A pointer to these results is now included in the main text. The complexity AgraSSt mainly stems from two sources,
> computing the graph kernel and computing the re-sampled Stein operator of size $B$. Computing the graph kernel depends on the choice of kernel and is not a focus of the proposed algorithm. The re-sampling procedure makes AgraSSt more scalable, as not all potential edges need to be used. The re-sampling algorithm has runtime $B^2$.
>
>
> Another weakness on your list is that the method description is quite dense and suffers from ``mathiness''. We have tried to make it the arguments easy to follow and have streamlined the definition of an ERGM further, but for providing theoretical underpinnings of the procedure, in order to make the paper self-contained, considerable notation was required. We would be happy to accept suggestions about how to simplify the presentation further without loss of accuracy.
>
> Addressing your questions:
> * Eq.(4) had indeed a typo which is now fixed. We apologize for it and realise that this may have made the paper harder to read.
>
> * The constraints on graph statistics in AgraSSt which are required for the theoretical justification is that $q_{\underline{k}} (x)>0$ for all $x$ so that the statistic has a positive probability to be present in all networks. The requirement is guided by the fact that the empirical Stein operators characterise the conditional distributions. If the conditional distributions were trivial then the AgraSSt procedure would not be meaningful.
>
>
> * In Section 5.2.3 the ''Edges'' statistic is used; this is the edge density after removing the respective vertex pair.
>
> * The conclusion in Section 5.2.3  between the difference between a single large component and two fairly separated communities was reached by visual inspection. This is now clarified in the new version.
>
> * As with any test statistic, we cannot be sure that we have not missed important graph statistics. Importance will relate to the research question (such as classifying edges or assessing homophily) which can be addressed in downstream analysis.
>
> Regarding the limitations, it is not the case that limitations are not discussed. In Section 6, it is indicated that AgraSSt currently only works for undirected unweighted graphs and does not include exogenous features. It is also clearly stated that ''As AgraSSt depends on the chosen summary statistic $t(x)$, results have to be interpreted with regards to the respective conditional distribution.''
>
> We hope that this reply has addressed your concerns.

---

> ### Comment · Reviewer_QFs7 · 2022-08-05
> **Response to rebuttal**
>
> Thank you for the thorough response to my comments. The authors have adequately addressed most of my concerns. Thus, I have increased my score accordingly and am in favor of acceptance.

---

> > ### Author Response · Authors · 2022-08-05
> > **response**
> >
> > Thank you very much for the update. We are very pleased  that we have addressed your main concerns and you now support accepting it!

---

### Official Review · Reviewer_Rhfm · 2022-07-10

**Rating:** 7
**Confidence:** 4
**Soundness:** 3 good
**Presentation:** 4 excellent
**Contribution:** 3 good

**Summary:**

This manuscript proposed a novel statistical procedure coined AgraSSt to assess the quality of implicit graph generators. AgraSSt provides interpretable criticisms for a graph generator training procedure. The manuscript gives theoretical guarantees and provide extensive empirical evaluations.

**Questions:**

How is the consistency of $\widehat q_t$ in theorem 3.2 verified ? The reviewer didn't find out results addressing the algorithm for estimating $\widehat q_t$.

**Limitations:**

The author adequately addressed the limitations and potential negative societal impact of their work.

**Strengths And Weaknesses:**

Strength: The idea of using Stein Statistics to construct rejection region for implicit models that widely presents in the literature is novel. The reviewer believed this idea has further implications that has also been well addressed in the future work part of the manuscript. The extensive empirical evaluation justified that the AgraSST is meritable on the real-world data application. Since the major contribution is the methodology, I believed the theoretical analysis is sufficient.

Weakness: Minor weakness which is discussed in the next section.

---

> ### Author Response · Authors · 2022-08-01
> **Consistent estimation of the conditional probability $\hat q_t$**
>
>
> Thank you for your assessment.
> To address your question about how the consistency of $\hat{q}_t$ in Theorem 3.2 is verified:
>
>
> In general, a look-up table can be used to estimate the conditional probabilities; generating many independent copies from the synthetic graph generator, consistency follows from the law of large numbers: we take the proportion of how often the edge s is present in networks for which $t(x_{-s})$ equals $k$. When using smoothing to estimate the conditional probability, consistency will depend on the smoothing method.  This is now also clarified in the new version. For edge-exchangeable networks, the consistency of $\hat q_t$ in Theorem 3.2 is addressed in Proposition A.1 in  SI.A. A corresponding pointer has been added to the new version.

---

> > ### Comment · Reviewer_Rhfm · 2022-08-08
> > **Response to the author**
> >
> > The reviewer thank the response from the author is sufficient and maintain his initial assessment of this work. Also, the reviewer apologise for the latency of response.

---

> > > ### Author Response · Authors · 2022-08-08
> > > **Response**
> > >
> > > Thank you for your update.

---

### Official Review · Reviewer_dJCC · 2022-07-10

**Rating:** 6
**Confidence:** 5
**Soundness:** 3 good
**Presentation:** 2 fair
**Contribution:** 3 good

**Summary:**

This paper introduces a statistical procedure for assessing the quality of graph generators. The procedure, namely, AgraSSt, approximates the Stein operator on implicit graph generators. This is done inspired by to the graph kernel Stein statistic (gKSS) for goodness-of-fit test on random graphs. The key step in this process is to estimate the conditional probability of an edge $s$ being present conditional on $t(x_{−s})=k$ by an estimator $\hat g_t (s; k)$; where t(x) denotes a statistic on graphs. AgraSSt is evaluated using synthetic data ERGM (specifically the Edge- 2Star-Triangle (E2ST) model) and compared with 5 possible graph characteristics (degree, total variation, Mahalanobis distance between degree distributions, via a parametric model, and gKSS). AgraSSt was also evaluated on real datasets to test the goodness-of-fit of various graph generators trained using Zachary’s Karate Club  network. The generators considered were: GraphRNN, NetGAN, CELL, and Monte Carlo network sampling.

**Questions:**

Using graph statistics to define AgraSSt provides flexibility. However, how can this technique provide adequate assessment for graph communities?
These are statistics that many degrees of freedom not easily solved by techniques driven by graph-statistics. How can AgraSSt deal with this situations?

More than by Stein operators for random graphs (as stated in the abstract), AgraSSt is inspired by graph kernel Stein statistics (gKSS). What is the main take-away you can describe from AgraSSt that is not present in gKSS?

**Limitations:**

Limitations are discussed with respect to the type of interpretations due to multiple-test correction that can affect areas such as personal health. Limitations of the technique presented are also discussed from a methodological perspective in Section 6.

**Strengths And Weaknesses:**

In summary, the paper is somewhat clear but more could be added to make it self-contained. The organization is not bad. The originality is mostly about the specific equality constrain considered during sampling. The quality of the work is theoretically good and experimentally reasonable. Some specific details include:

Strengths:
The significance of a possible solution to the problem of quality assessment of implicit graph generators is profound. This could have impact to validate models in various domains in science where GNN models are used to capture relational structure in data.
The theoretical analysis seems sound and can justify the consistency of the AgraSSt estimator.
The paper presents validation of the estimator using various graph statistic (not just one) using synthetic data and compares the graph assesment using various graph generators.


Weaknesses:
The paper relies in concepts that are not thoroughly introduced in the paper and thus, the paper lack self-containment. For instance what is the effect of exchangeability, why is this important in non-implicit graph generators?
The experiments are interesting but only provide an exploratory perspective of the graph generator assessment tool. Since there are no other alternatives to compare with, it may important to use some basic naive baselines to do so. For instance, by modifying an existing generator to build empirical distributions that could be used for hypothesis testing.

---

> ### Author Response · Authors · 2022-08-01
> **Clarity, empirical comparison and nonparametric model assessment**
>
>
> We are pleased that you find  ''The organization is not bad'' and  ''The quality of the work is theoretically good and experimentally reasonable''
> but still your overall assessment is ''3: Reject: For instance, a paper with technical flaws, weak evaluation, inadequate reproducibility and incompletely addressed ethical considerations.''
>
> From your report we could not quite understand how you reached this conclusion.
>
> You point out as a weakness that the paper relies on concepts which are not thoroughly introduced in this paper. We have tried to make sure that every concept is introduced. The edge-exchangeability is not a requirement for Theorem 3.2, our main theoretical result. However estimating the conditional probabilities is easier under edge-exchangeability, as is shown in Proposition A.1. We have added some text in the new version to make this clearer.
>
> The experiments illustrate the theoretical results. As detailed in Section 5.2.2, we use the standard Monte Carlo network sampling method from the $\texttt{ergm}$ R package as a baseline to compare against, regarding generating samples. Moreover, we compare against the standard graphical tests from the $\texttt{ergm}$ R package, see Figure 7 in SI D.
>
> Regarding your specific questions:
>
> ``How can AgraSSt be used to provide adequate assessment for graph communities?''AgraSSt is designed to assess synthetic data generators, not to pick up particular features of the observed network. Once a synthetic data generator has passed the assessment, one can generate samples from it and use these to study features of the original network, in a downstream analysis which is not carried out in the paper. In the Karate network example, Fig.5 in SI D, if the interest lies in community detection, one could calculate the modularity of the accepted and rejected samples and compare them to the modularity of the Karate network. This analysis would point to the modularity being an important feature of the network. Trying this out the modularity for the Karate club and for the average based on 8 samples each from the graph generators is
> given in the following table.
>
> |    Karate Club Data    | GraphRNN | NetGAN | CELL | Original network |
> | ----------- | ----------- | ----------- |----------- |----------- |
> | mean modularity ($\pm$std) | 0.4104 ($\pm$0.0357) | 0.3628 ($\pm$0.0479) | 0.3668 ($\pm$0.0177) | 0.3806  |
>
>
>
> The CELL generator, which performed best in AgraSSt, has the modularity which is closest to the observed modularity. Moreover, it has by far the smallest standard deviation.
>
>
> ``What is the main takeaway you can describe from AgraSSt that is not present in gKSS?'' The gKSS method is parametric and developed for exponential random graph models and can only be applied to assess the goodness of fit for exponential random graph models. AgraSSt is nonparametric and can be applied to any random graphs generator.
>
>
> We hope that this reply has alleviated your concerns.

---

> > ### Comment · Reviewer_dJCC · 2022-08-08
> > **Response**
> >
> > Thank you, this clarifies most of my concerns. I will raise my score in consequence. However, my comment with respect to the paper not been totally self-contain remains. That is the concept of edge-exchangeability for non-implicit graph generators (Thm 3.3 of the paper) is not clear. Please, include these clarifications in the main text of the paper.

---

> > > ### Author Response · Authors · 2022-08-08
> > > **Further clarification on edge-exchangeability in the main text**
> > >
> > > Many thanks for your revision of your score and for your helpful suggestion. We may have misunderstood before what you meant. In our latest version, we have now clarified the definition of edge-exchangeable graphs just before Theorem 3.3, to read
> > >
> > >
> > > ''In SI.A we prove the following result for edge-exchangeable graphs, that is, graphs in which any
> > > permutation of the edge indicators have the same distribution.’’
> > >
> > >
> > >
> > > We hope that this change makes the paper more self-contained along the lines which you have suggested.

---

### Official Review · Reviewer_MKcL · 2022-07-11

**Rating:** 7
**Confidence:** 3
**Soundness:** 3 good
**Presentation:** 3 good
**Contribution:** 3 good

**Summary:**

This paper proposes a novel statistical procedure to evaluate and interpret implicit graph generators based on approximate graph Stein statistics.

**Questions:**

Questions:

- What is the effect of $\beta_2$ (equation (11)) in sparsity? Does sparsity grow or decrease from left to right in Figures 1a-1c?
- How was the hyperparameter selection done for the GraphRNN, NetGAN, and CELL in Table 1?

Minor comments:

- Typo in line 72 (comma before For).
- The authors should explicitly write the assumptions/conditions necessary for stating Lemma 3.1 in the body of the lemma (instead of writing "In this setting, ...").
- For readability, the update equations in lines 137-138 and equation 8 should be included in Algorithm 1. "be" missing in instruction 2 of Algorithm 1.
- Line 162 should be an equation.
- Better explain the axes of Figures 1a-1c.
- Include a more precise description of the key (colors) of Figure 1c.


**Strengths And Weaknesses:**

Strengths:

- The paper is well-motivated and well-written. It is also timely, as it echoes the efforts of the community towards more explainable machine learning models.
- The numerical results comparing AggraSSt with other methods are convincing, and show that AgraSST outperforms them empirically.

Weaknesses:

- The fact that AgraSSt depends on the chosen summary statistic $t(x)$ is not necessarily a weakness, but it requires expert selection. This should be more stressed throughout the paper. The effects of a specific (good or poor) choice of $t(x)$ should also be illustrated.
- It is not clear whether AgraSSt does better than, e.g., only comparing sums of degrees for vertex pairs $s$ (Sum_deg in Figure 1), simply because $t(x)$ encompasses more summary statistics. In other words, does AgraSSt bring any intrinsic advantages, or do they only come for the fact that it allows to simultaneously consider multiple summary statistics?
- It would have been interesting to see numerical experiments in graphs with variable sparsity levels (e.g., power law, expander graphs), and possible failure cases for AgraSSt (perhaps related to the choice of $t(x)$).

---

> ### Author Response · Authors · 2022-08-02
> **Choices of graph statistics, sparsity and several clarifications**
>
> Thank you for your assessment that the paper is well-motivated, well-written and timely.
>
> Regarding the fact that AgraSSt depends on the choice of summary statistic, indeed some care has to be taken in selecting them. A brief discussion is given in Section 3.3 and SI D.3. We have now flagged this further in the Discussion, Section 6, to read ''nvestigating the effect of choice of $t(x)$ will be part of future work.''
>
> You ask whether AgraSSt brings any intrinsic advantages, or do the advantages only come for the fact that it allows to simultaneously consider multiple summary statistics. We find that the simple ``Edges'' statistic which for every vertex pair $s$ counts the number of edges in the graph with $s$ left out, does better than more complex summary statistics, see Section 5.1.2. This summary statistic is no more complex than the ones used for MDdeg pr TV\_deg, as shown in Figure 2. This indicates that the superior performance of AgraSSt is not due to the fact that multiple summary statistics could be considered.
>
> To address your question regarding sparsity, and also your question regarding the effect of $\beta_2$ on sparsity:
>
> An experiment with varying sparsity levels is shown in Figure 1.
> The sparsity varies with the coefficient $\beta_2$ for the $2$-star statistic. The  larger $\beta_2$, the sparser the graph becomes on average, as $\beta_1$ and $\beta_3$ in Eq.(11) are fixed. In particular, when the coefficient becomes negative then the alternative model is sparser than the null model; and when the coefficient is larger than $0$, the alternative model is denser than the null model.
> We agree that more synthetic experiments regarding heterogeneity and varying levels of sparsity could be of interest, and this will be part of future work.
>
> For a theoretically justified choice of graph statistics, we require $q_\underline{k} (x) > 0$ for all $x$, that is, we need a statistic which is present in all networks, see Lemma 3.1. As a simple example that such condition may fail, we could use for $s = (i,j)$ the number of triangles which involve $i$ and $j$ and take as network generator a BA network generator with $m=1$, which does not contain any triangles. In this case, such a triangle statistic is considered a ''bad'' choice of graph statistic.
>
> You also ask about how the hyperparameter selection was carried out. The hyperparameters are chosen from the implementation suggested in the respective original papers; we checked that the training loss decreased until convergence. As our goal is to understand the model performance, these hyperparameters are not extensively tuned.
>
> For GraphRNN, we use batch size 128, epoch 1000 for training, 100 for testing, and learning rate 0.003.
> For CELL, we use a learning rate 0.01, and weight decay 1e-7.
> For NetGAN, we use batch size 128, epoch 50, generator size and discriminator size both 128, and learning rate 0.0003. These settings have now been added in SI D.6.
>
> Following your suggestion, we have now included the assumptions in Lemma 3.1 explicitly.
>
> In Algorithm 1 we deliberately did not include Eq.(8) as that is a particular possibility for estimating the conditional distribution which is valid when the underlying graph has exchangeable edge indicators. We feel that including Eq.(8) in the algorithm may be too restrictive.
>
> Figure 2 is now better annotated.
>
> Finally, we would like to thank you for spotting the typos, which have been amended.
>
> We hope that this reply and the changes in the paper have addressed your concerns in a satisfactory way.

---

> > ### Comment · Reviewer_MKcL · 2022-08-08
> > **Comments addressed, score increased**
> >
> > Thank you for your detailed response. All of my comments and questions have been addressed, so I'm increasing my score to 7.

---

> > > ### Author Response · Authors · 2022-08-08
> > > **Reply**
> > >
> > > Thank you very much for the update. We are very pleased that we have addressed your questions and concerns. And thanks for the score update!

---

### Meta-Review · Area_Chair_dind · 2022-08-26

**Recommendation:** Accept
**Confidence:** Certain

**Metareview:**

After the author response period, the reviewer ratings were all positive. The reviewers felt that the paper tackles an important problem in assessing the quality of graph generators and proposes a novel, general purpose, and effective approach.

The reviewers pointed out several points for clarification in their reviews, which we hope that the authors will address in the final version of the paper. Many of these have been addressed during the author response period.

**Award:**

No

---

### Decision · Program_Chairs · 2022-09-14

Accept